# Clueless/CLUH regulates mitochondrial fission by promoting recruitment of Drp1 to mitochondria

Huan Yang ⬡ [1], Caroline Sibilla[2,3,9], Raymond Liu[4,10], Jina Yun[1,11], Bruce A. Hay[4], Craig Blackstone ⬡ [2,12], David C. Chan[4], Robert J. Harvey ⬡ [5,6] & Ming Guo ⬡ [1,7,8✉]

Mitochondrial fission is critically important for controlling mitochondrial morphology, function, quality and transport. Drp1 is the master regulator driving mitochondrial fission, but exactly how Drp1 is regulated remains unclear. Here, we identified *Drosophila* Clueless and its mammalian orthologue CLUH as key regulators of Drp1. As with loss of *drp1*, depletion of *clueless* or *CLUH* results in mitochondrial elongation, while as with *drp1* overexpression, *clueless* or *CLUH* overexpression leads to mitochondrial fragmentation. Importantly, *drp1* overexpression rescues adult lethality, tissue disintegration and mitochondrial defects of *clueless* null mutants in *Drosophila*. Mechanistically, Clueless and CLUH promote recruitment of Drp1 to mitochondria from the cytosol. This involves CLUH binding to mRNAs encoding Drp1 receptors MiD49 and Mff, and regulation of their translation. Our findings identify a crucial role of Clueless and CLUH in controlling mitochondrial fission through regulation of Drp1.

[1] Department of Neurology, UCLA David Geffen School of Medicine, Los Angeles, CA, USA. [2] Cell Biology Section, Neurogenetics Branch, National Institute of Neurological Disorders and Stroke, Bethesda, MD, USA. [3] Department of Pharmacology, University College London School of Pharmacy, London, UK. [4] Division of Biology and Biological Engineering, California Institute of Technology, Pasadena, CA, USA. [5] School of Health and Behavioural Sciences, University of the Sunshine Coast, Sippy Downs, QLD, Australia. [6] Sunshine Coast Health Institute, Birtinya, QLD, Australia. [7] Department of Molecular and Medical Pharmacology, UCLA David Geffen School of Medicine, Los Angeles, CA, USA. [8] California NanoSystems Institute at UCLA, Los Angeles, CA, USA. [9] Present address: AstraZeneca PLC, Cambridge Biomedical Campus, Cambridge, UK. [10] Present address: Department of Microbiology and Immunology, UCSF, San Francisco, CA, USA. [11] Present address: Genentech, Inc., South San Francisco, CA, USA. [12] Present address: Department of Neurology, Massachusetts General Hospital, Harvard Medical School, Boston, MA, USA. ✉email: mingfly@ucla.edu

Mitochondria are dynamic organelles, and their morphology and function are governed by a balance between the opposing actions of fusion and fission[1]. Mitochondrial fusion is controlled by Mitofusin (Mfn) proteins[2] and Opa1[3], and mitochondrial fission is driven by Drp1[4]. Mitochondrial fission plays pivotal roles in cell division[5], regulated cell death[6], mitochondrial quality control through mitophagy[7,8], and mitochondrial DNA inheritance[9,10]. At the organismal level, dysregulation of mitochondrial fission leads to disrupted metabolic homeostasis[11], defective brain development[12], cardiovascular diseases[13], and cancer[14]. Drp1 dysfunction has also been implicated in the pathogenesis of several neurodegenerative diseases, including Alzheimer's disease, Parkinson's disease (PD), amyotrophic lateral sclerosis, and Huntington's disease[15–17]. With respect to PD, we and others have shown that *PINK1* and *parkin*, mutations that lead to recessive forms of PD[18,19], function in the same pathway to regulate mitochondrial integrity[20–24] and mitophagy[25–27]. Overexpression of *drp1* rescues mitochondrial and tissue defects of *PINK1* or *parkin* mutants in *Drosophila*[22,23]. Drp1 also facilitates segregation of the damaged parts of mitochondria, which are targeted for mitophagy mediated by PINK1 and Parkin[7,8,25].

Drp1 is a member of the dynamin superfamily of GTPases. It is localized predominantly in the cytosol and is recruited onto mitochondria via interactions with Drp1 receptors anchored to the outer mitochondrial membrane (OMM). In mammalian cells, these receptors include mitochondrial fission factor (Mff) and mitochondrial dynamics proteins of 49 and 51 kDa (MiD49 and MiD51)[28–32]. Once recruited onto the OMM, Drp1 co-assembles with these receptors, forming an oligomeric ring to constrict mitochondria and drive fission[32].

*clu* gene orthologues can be found in evolutionarily distant eukaryotes, including yeast[33], ameba[34,35], *Arabidopsis*[36], *Drosophila* (*clueless*, referred to as "*clu*" hereafter)[37,38] and mammals (*CLUH*)[39,40]. In all of the above species, cells with loss of *clu* orthologues show a phenotype of clustered mitochondria in the perinuclear region, in contrast to wild-type cells in which mitochondria are dispersed throughout the cytoplasm. However, the mechanism underlying this phenotype remains unclear. Clu orthologues have been identified as RNA-binding proteins in yeast[41], *Drosophila*[42], and mammalian cells[39,43]. Clu orthologues also share an evolutionarily conserved domain structure[37], with a highly conserved N-terminal Clu domain, and a C-terminal tetratricopeptide repeat (TPR) domain that is responsible for binding RNAs[42]. While primarily cytoplasmic, *Drosophila* Clu protein is found in granules juxtaposed with mitochondria in female germline cells. These are thought to be ribonucleoprotein (RNP) complexes formed as an adaptation to metabolic changes[37,44]. In mammalian cells, CLUH specifically binds multiple mRNAs of nuclear-encoded, mitochondria-destined proteins that belong to critical metabolic pathways, such as oxidative phosphorylation (OXPHOS), tricarboxylic acid (TCA) cycle, and fatty acid oxidation[39,45]. CLUH is distributed throughout the cytosol under unstressed conditions. In contrast, during starvation, CLUH and its bound mRNAs form large RNP particles that function as compartments within which CLUH regulates translation and stability of these mRNAs, controls mTORC1 signaling, and modulates metabolic rewiring[39,45,46]. In addition, Clu associates with mitochondrial proteins, including TOM20, Porin, and PINK1, as well as mitochondrially localized Parkin following mitochondrial damage[47]. Importantly, *clu* interacts genetically with *PINK1* and *parkin* in *Drosophila*, with previous reports showing that overexpression (OE) of *clu* rescues *PINK1* but not *parkin* mutant phenotypes[47,48].

In this study, we identify *Drosophila* Clu and human CLUH as key upstream regulators of Drp1 to control mitochondrial fission. We show that Clu and CLUH promote the recruitment of Drp1 onto mitochondria from the cytosol. This involves CLUH binding to mRNAs encoding Drp1 receptors Mff and MiD49 and regulation of their translation. In addition, we found that similarly to *drp1* OE, *clu* OE rescues mitochondrial and tissue defects of *parkin* null mutants, in addition to *PINK1* null mutants, in *Drosophila*. Our study provides new mechanistic insights into how Drp1 activity and mitochondrial fission are regulated.

## Results

### *clu* OE suppresses, and *clu* loss-of-function exacerbates, *PINK1* and *parkin* null mutant phenotypes in *Drosophila*. We and others first demonstrated that *PINK1* and *parkin* function in a common pathway to regulate mitochondrial integrity and quality in *Drosophila*[20,21]. *Drosophila* muscle contains organized, high-density mitochondria that fill the spaces between myofibrils. Loss of *PINK1* or *parkin* leads to severe mitochondrial and tissue defects, including thoracic indentation that reflects underlying muscle degeneration (Supplementary Fig. 1a–c), cell death (compare Fig. 1a to Fig. 1b, d), and disrupted tissue integrity (compare Fig. 1a' to Fig. 1b', d'). Previous work has suggested that OE of *clu* rescues *PINK1* null but not *parkin* null mutant phenotypes[47,48]. In contrast, by using the strong muscle driver Mef2-GAL4, we found that OE of *clu* suppressed all the above defects in *parkin* null mutants, in addition to *PINK1* null mutants (Fig. 1c, c', e, e', quantified in g, h; Supplementary Fig. 1a–c). Using the same GAL4 driver, we previously showed that *drp1* OE similarly suppresses phenotypes due to loss of either *PINK1* or *parkin*[22,23].

*PINK1* null and *parkin* null mutants are both viable, as are double null mutants[20,24]. *clu* null mutants are also viable, though they have a greatly reduced lifespan, dying 3–6 days after eclosure[37]. In contrast, we found that double null mutants of *PINK1 clu* and *parkin clu* were both lethal, with death during late pupal stages. The lethality of *parkin clu* double null mutants has also been noted previously[48]. These synthetic lethal interactions are reminiscent of previous work by us and others, which showed that loss of *drp1* function in *PINK1* or *parkin* null mutant background also results in lethality in *Drosophila*[22,23].

To bypass the lethality issue in order to study phenotypes of *PINK1 clu* and *parkin clu* double mutants, we utilized a *clu* partial loss-of-function (*clu^d00713^* hypomorphic) mutant in the *PINK1* or *parkin* null mutant background. With reduced Clu protein levels[37], *clu^d00713^* mutants did not show a greatly reduced lifespan[37] or muscle disintegration (Supplementary Fig. 1d, e), in contrast to *clu* null mutants. *clu^d00713^* mutants showed more elongated mitochondria (Fig. 1i, j, quantified in o; Supplementary Fig. 1d–d", e–e", quantified in f), as with *PINK1* or *parkin* null mutants (Fig. 1k, l). Strikingly, *PINK1 clu^d00713^* (Fig. 1m) and *parkin clu^d00713^* double mutants (Fig. 1n-n') showed highly elongated, interconnected, and enlarged mitochondria, which were much more severe than any single mutant alone (Fig. 1j–l, quantified in o). In addition, while *clu^d00713^* mutants showed ATP levels comparable to those in wild-type flies, *PINK1 clu^d00713^* and *parkin clu^d00713^* double mutants showed a greater decrease in ATP levels than was observed in either *PINK1* or *parkin* null mutant alone (Fig. 1p). Therefore, like *drp1*, *clu* OE suppresses, and *clu* loss-of-function exacerbates, *PINK1* and *parkin* null mutant phenotypes in *Drosophila* (Fig. 1q). Together, these results show that *clu* acts in parallel to the *PINK1-parkin* pathway to regulate mitochondrial morphology and function.

### *clu* regulates mitochondrial morphology in *Drosophila*. To investigate if *clu* suppresses *PINK1* and *parkin* null mutant phenotypes by regulating mitochondrial fission or fusion, we

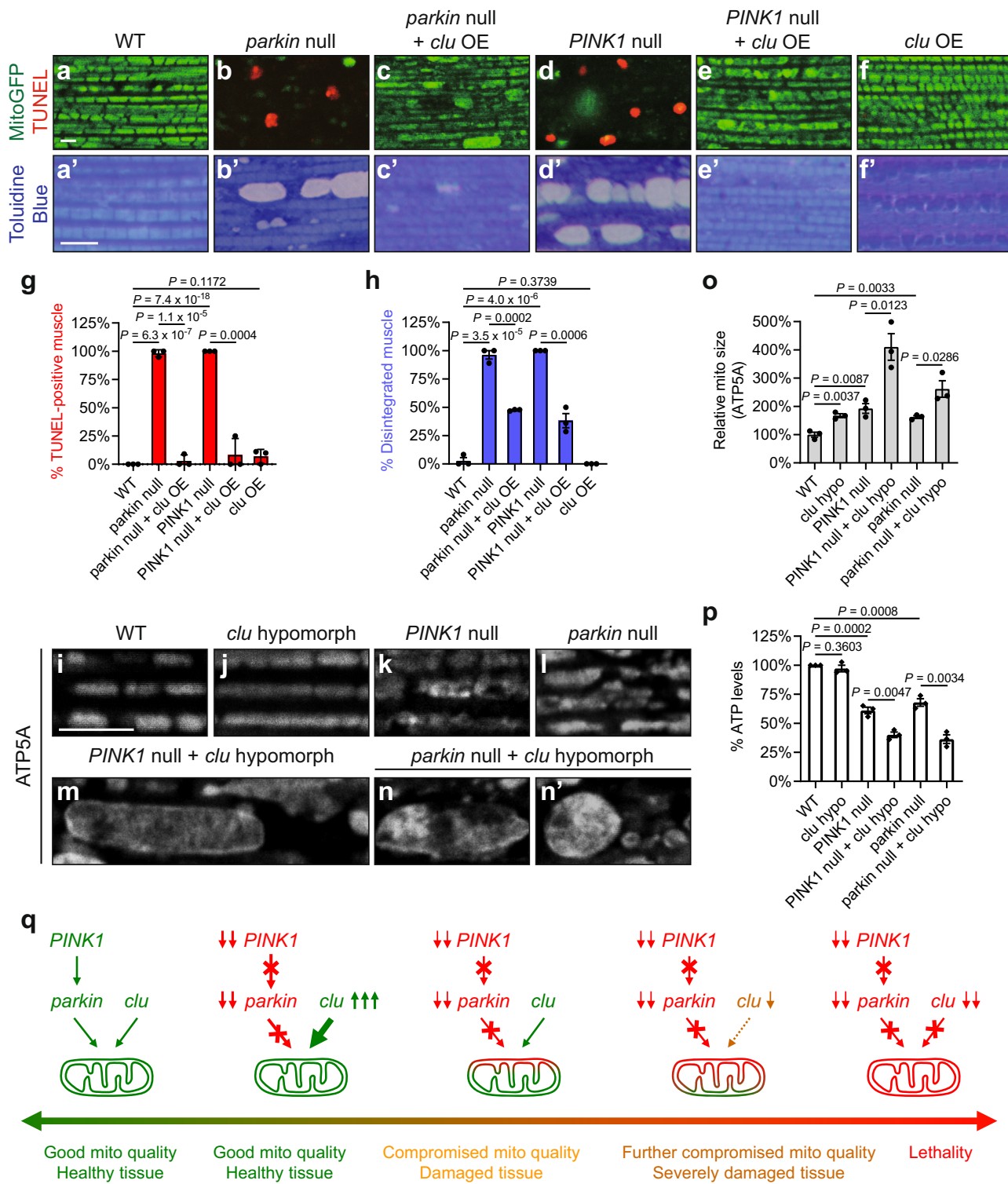

examined mitochondrial morphology in response to muscle-specific *clu* RNAi[48] or *clu* OE. *clu* RNAi resulted in significantly elongated mitochondria, as visualized by mitoGFP (Fig. 2a, b) and with transmission electron microscopy (TEM) (Fig. 2d–d', e–e'). Mitochondrial elongation was also seen in *clu* hypomorphic mutants as shown above (Fig. 1j, o; Supplementary Fig. 1e–e", f). *clu* OE led to mitochondrial fragmentation as visualized by mitoGFP (Figs. 1f and 2c). Under TEM, muscle with *clu* OE had mitochondria that were smaller in size and with uneven cristae

density (Fig. 2f–f'). Changes in mitochondrial morphology are statistically significant (Fig. 2g, h). These results show that *clu* regulates mitochondrial morphology in a pro-fission manner as *drp1* does.

**Overexpression of *drp1* suppresses adult lethality, tissue damage, and mitochondrial defects of *clu* null mutants in *Drosophila*.** To further explore whether *clu* regulates

**Fig. 1 *clu* overexpression suppresses, and *clu* loss-of-function exacerbates, *PINK1* and *parkin* null mutant phenotypes in *Drosophila*. a–f** Confocal microscopy images of the thoracic muscle labeled with mitoGFP (green) and TUNEL (red). **a′–f′** Toluidine Blue staining of plastic sections of embedded thoraces. *clu* overexpression (OE) suppresses muscle death (**a–f**) and tissue disintegration (**a′–f′**) in *parkin* null mutants (*park²⁵/dpk²¹*), in addition to *PINK1* null mutants (*PINK1⁵*). Expression of UAS-mitoGFP (**a–f**) and UAS-*clu* (**c, c′, e, e′, f, f′**) were driven by Mef2-GAL4. **g, h** Quantification of muscle death (**g**) and tissue disintegration (**h**). For each genotype, 3 male flies were analyzed (mean ± SEM, *n* = 3). For each fly, 30–50 muscle pieces were analyzed, and the percentage of TUNEL-positive muscle (**g**) or that of disintegrated muscle (**h**) was calculated. **i–n′** Mitochondria in the muscle are labeled using a mouse anti-ATP5A antibody. Mitochondria are more elongated in *clu^d00713^* homozygotes (**j**), as compared to those in wild-type (WT) flies (**i**). *PINK1⁵* (**k**) and *park²⁵/dpk²¹* (**l**) single mutants show elongated mitochondria with vacuolation. *clu^d00713^* homozygotes in the *PINK1⁵* or *park²⁵/dpk²¹* background show exacerbated mitochondrial defects, including enlargement, severe vacuolation, as well as irregular shape and distribution (**m–n′**). **o** Quantification of the relative mitochondrial sizes in (**i–n′**). For each genotype, 3 male flies were analyzed (mean ± SEM, *n* = 3). For each fly, 30–50 mitochondria were analyzed using Fiji/ImageJ and the average mitochondrial size was calculated. **p** Results of ATP measurements using whole fly lysates. *clu* hypo (hypomorph): *clu^d00713^* homozygotes. Experiments were performed in triplicate (mean ± SEM, *n* = 3). **q** A schematic illustration of the genetic interactions between *clu* and the *PINK1–parkin* pathway in *Drosophila*. *PINK1* and *parkin* function in the same pathway to regulate mitochondrial integrity. Loss of either *PINK1* or *parkin* results in severe mitochondrial dysfunction and tissue damage. *clu* overexpression suppresses phenotypes due to loss of *PINK1* or *parkin*. Partial loss of *clu* function (*clu^d00713^* mutants) exacerbates either *PINK1* null or *parkin* null mutant phenotypes. Complete loss of *clu* function (*clu^f04554^* mutants) in *PINK1* or *parkin* null mutant background results in lethality. **a–f′, i–n′** Scale bars: 5 µM. **g–o, p** One-way ANOVA with post hoc Tukey's HSD test.

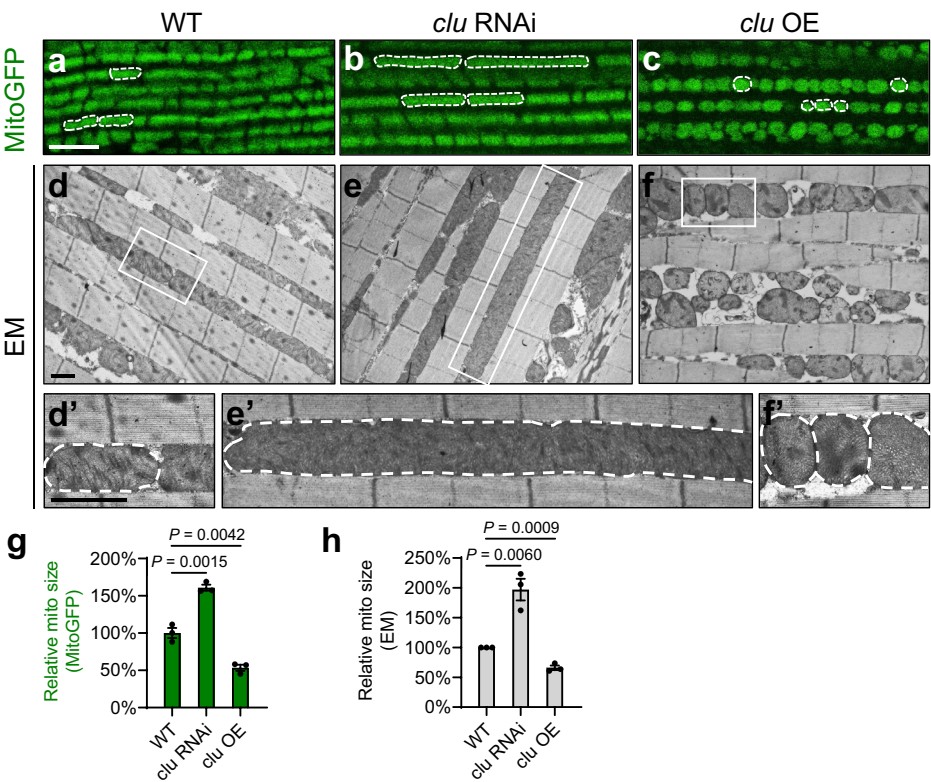

**Fig. 2 *clu* regulates mitochondrial morphology in *Drosophila*. a–c** Confocal microscopy images of the thoracic muscle labeled with mitoGFP. *clu* RNAi leads to mitochondrial elongation (**b**), and *clu* overexpression (OE) results in mitochondrial fragmentation (**c**), as compared with the wild type (WT, **a**). White dashed lines mark the boundaries of representative mitochondria. Scale bar: 5 µM. **d–f, d′–f′** TEM ultrastructural images show that mitochondria in wild-type muscle are well aligned in between myofibrils (**d, d′**). Muscle with *clu* RNAi exhibits dramatically elongated mitochondria (**e, e′**), and muscle with *clu* overexpression shows fragmented mitochondria as well as cristae with uneven density (**f, f′**). **d′–f′** Enlarged images of the boxed regions in (**d–f**). White dashed lines illustrate the boundaries of representative mitochondria. Scale bars: 1 µM. Expression of UAS-mitoGFP (**a–c**), UAS-*clu* RNAi (**b, e, e'**), and UAS-*clu* (**c, f, f′**) were driven by the indirect flight muscle (IFM)-specific driver IFM-GAL4[24]. Expression of UAS-*clu* RNAi driven by Mef2-GAL4 results in adult lethality and severe mitochondrial defects (data not shown), which are similar to *clu^f04554^* mutants (see Fig. 3). **g, h** Quantification of the relative sizes of mitochondria labeled with mitoGFP (**g**) or visualized under TEM (**h**). For each genotype, 3 different male flies were analyzed (mean ± SEM, *n* = 3). For each individual fly, 30–50 individual mitochondria were analyzed using Fiji/ImageJ and the average mitochondrial size was calculated (one-way ANOVA with post hoc Tukey's HSD test).

mitochondrial morphology by inhibiting fusion or promoting fission, we investigated whether *clu* regulates *mfn* (also known as *Marf*) or *drp1*. *clu* null mutant flies are short-lived, dying 3–6 days after eclosure, with about 50% survival on Day 3, less than 25% survival on Day 4, about 5% survival on Day 5, and 0% survival after Day 6 (Fig. 3a, red line; Supplementary Tables 1 and

2). In contrast, wild-type flies can live over 100 days (Fig. 3a, black line). *clu* OE fully rescued the adult lethality of *clu* null mutants (Fig. 3a, blue line), confirming that the lethality of *clu* null mutants was indeed due to lack of *clu*. *mfn* RNAi did not rescue the lethality of *clu* null mutants. Strikingly, however, *drp1* OE significantly rescued the lethality due to lack of *clu* (Fig. 3a,

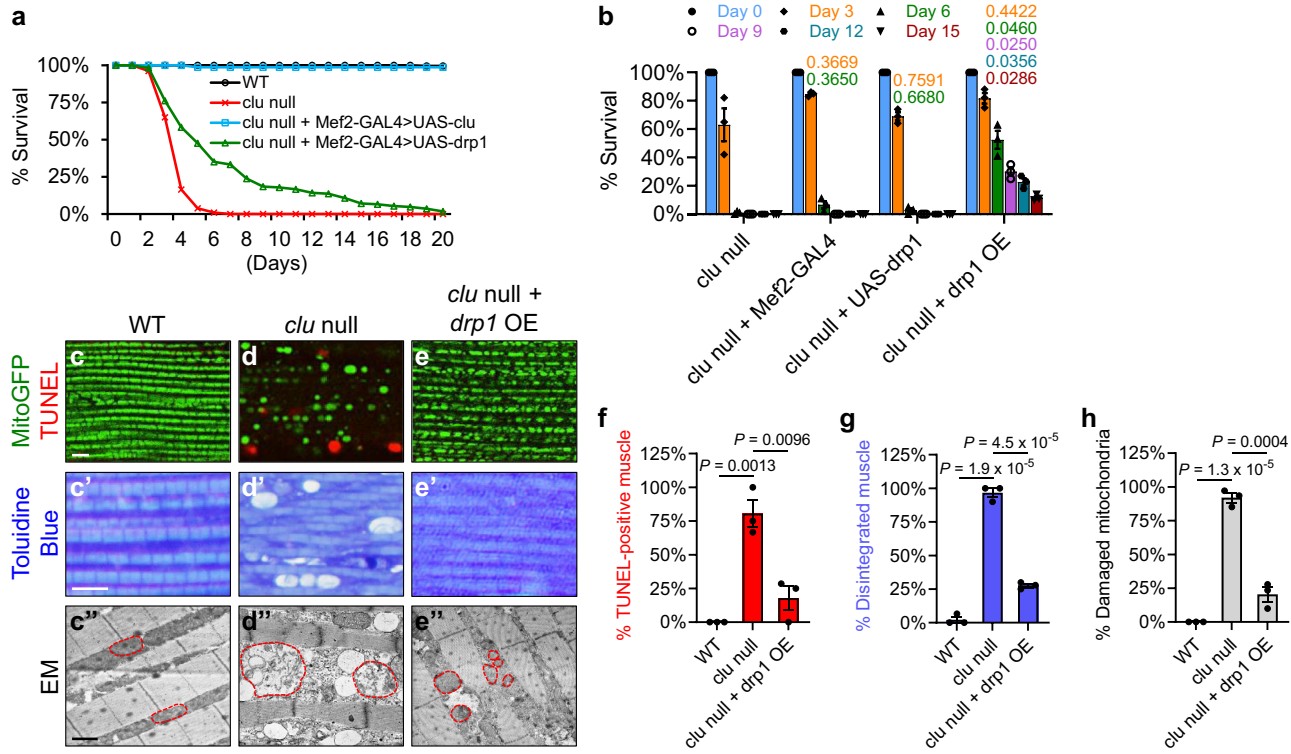

**Fig. 3 Overexpression of *drp1* suppresses *clu* null mutant phenotypes in *Drosophila*. a** Survival of wild-type (black line) and *clu^f04554* null mutant (red line) flies, as well as those with overexpression of *clu* (blue line) or *drp* (green line) in the *clu^f04554* background. Expression of UAS-*clu* and UAS-*drp1* were driven by Mef2-GAL4. Days with 50%, 25%, and 5% survival for these flies are shown in Supplementary Table 1. **b** Percentage of live flies on representative days. Statistical analysis of day-to-day survival rate is shown in Supplementary Table 2: overexpression of *drp1* significantly increases the survival rates of *clu^f04554* mutants throughout Day 4 to Day 16. As controls, the introduction of Mef2-GAL4 or UAS-*drp1* alone into the *clu^f04554* background does not result in significant differences in survival rates as compared to those of the *clu^f04554* mutants (one-way ANOVA with post hoc Tukey's HSD test, with *P*-values displayed in the graph. *P* < 0.05: significantly different from *clu^f04554* mutants). **c–e** Confocal microscopy images of the thoracic muscle double-labeled with mitoGFP (green) and TUNEL (red). **c′–e′** Toluidine Blue staining of plastic sections of embedded thoraces. **c″–e″** TEM ultrastructural images of thoracic muscle. Red dashed lines mark the boundaries of representative mitochondria. *drp1* overexpression suppresses TUNEL-positive signals indicative of cell death in *clu^f04554* mutants (**c–e**, quantified in **f**), largely restores tissue integrity (**c′–e′**, quantified in **g**) and mitochondrial cristae structure (**c″–e″**, quantified in **h**) in *clu^f04554* mutants, and results in small mitochondrial size (**e″**). Scale bars: 5 μM in (**c–e**, **c′–e′**), 1 μM in (**c″–e″**). Expression of UAS-mitoGFP (**c–e**) and UAS-*drp1* (**e**, **e′**, **e″**) were driven by Mef2-GAL4. **f–h** Quantification of muscle death (**f**), tissue disintegration (**g**), and damaged mitochondria with broken cristae (**h**). For each genotype, 3 different male flies were analyzed (mean ± SEM, *n* = 3, one-way ANOVA with post hoc Tukey's HSD test). **f**, **g** For each male fly, 30–50 individual muscle pieces were analyzed, and the percentage of TUNEL-positive muscle (**f**) or that of disintegrated muscle (**g**) was calculated. **h** For each male fly, 30–50 individual mitochondria were analyzed, and the percentage of vacuolated mitochondria with broken cristae was calculated.

green line): 50% survival was extended to Day 7 as compared with Day 3, 25% survival to Day 12 in contrast to Day 4, and 5% survival to Day 19 as compared with Day 5 (Supplementary Table 1). In addition, 53% of *clu* null mutants with *drp1* OE survived more than 6 days, 30% more than 9 days, 23% more than 12 days, and 11% more than 15 days, while all *clu* null mutants were dead by Day 7 (Fig. 3b; Supplementary Table 2).

*clu* null mutants have poor flight performance and a "held-up" wing phenotype[37], signs of indirect flight muscle defects[20]. Indeed, we observed cell death (Fig. 3d) and disrupted tissue integrity (Fig. 3d') in *clu* null mutant indirect flight muscle. Ultrastructural analysis using TEM revealed that *clu* null mutant muscle contained severely damaged mitochondria of uneven size, and many of these swollen mitochondria had few or no cristae (Fig. 3d", compared to the wild type in Fig. 3c–c"). Strikingly, when *drp1* was overexpressed in the muscle, cell death (Fig. 3e), disrupted tissue integrity (Fig. 3e') and swollen mitochondria (Fig. 3e") due to lack of *clu* were all suppressed (Fig. 3f–h), and cristae structure was restored (Fig. 3e"). Mitochondria in these flies were smaller, which was expected with *drp1* OE[22]. The striking rescue of adult lethality, tissue disintegration, and

mitochondrial defects in *clu* null mutants by *drp1* OE suggests that *clu* functions upstream to positively regulate *drp1*.

**clu regulates mitochondrial fission by promoting recruitment of Drp1 to mitochondria in *Drosophila*.** To further investigate the mechanism by which *clu* regulates *drp1*, we examined expression levels of Drp1 in response to *clu* loss-of-function or *clu* OE. To measure endogenous Drp1 protein levels, we used transgenic flies expressing a tagged *drp1* genomic transgene under the control of its endogenous promoter[49]. Western blotting results showed no significant changes in Drp1 protein levels in response to *clu* complete or partial loss-of-function, or *clu* OE, in the muscle (Fig. 4a, b). We then focused on *Drosophila* female germline and quantified Drp1 levels by immunofluorescence signals in individual nurse cells (Fig. 4c–e). Nurse cells grow extensively during oogenesis and become relatively large after the middle stage of oogenesis; these nurse cells offer a great resolution for examining protein levels. *clu* null mutants are female sterile because of developmental arrest at the middle stage of oogenesis[37], which makes it difficult to obtain nurse cells beyond

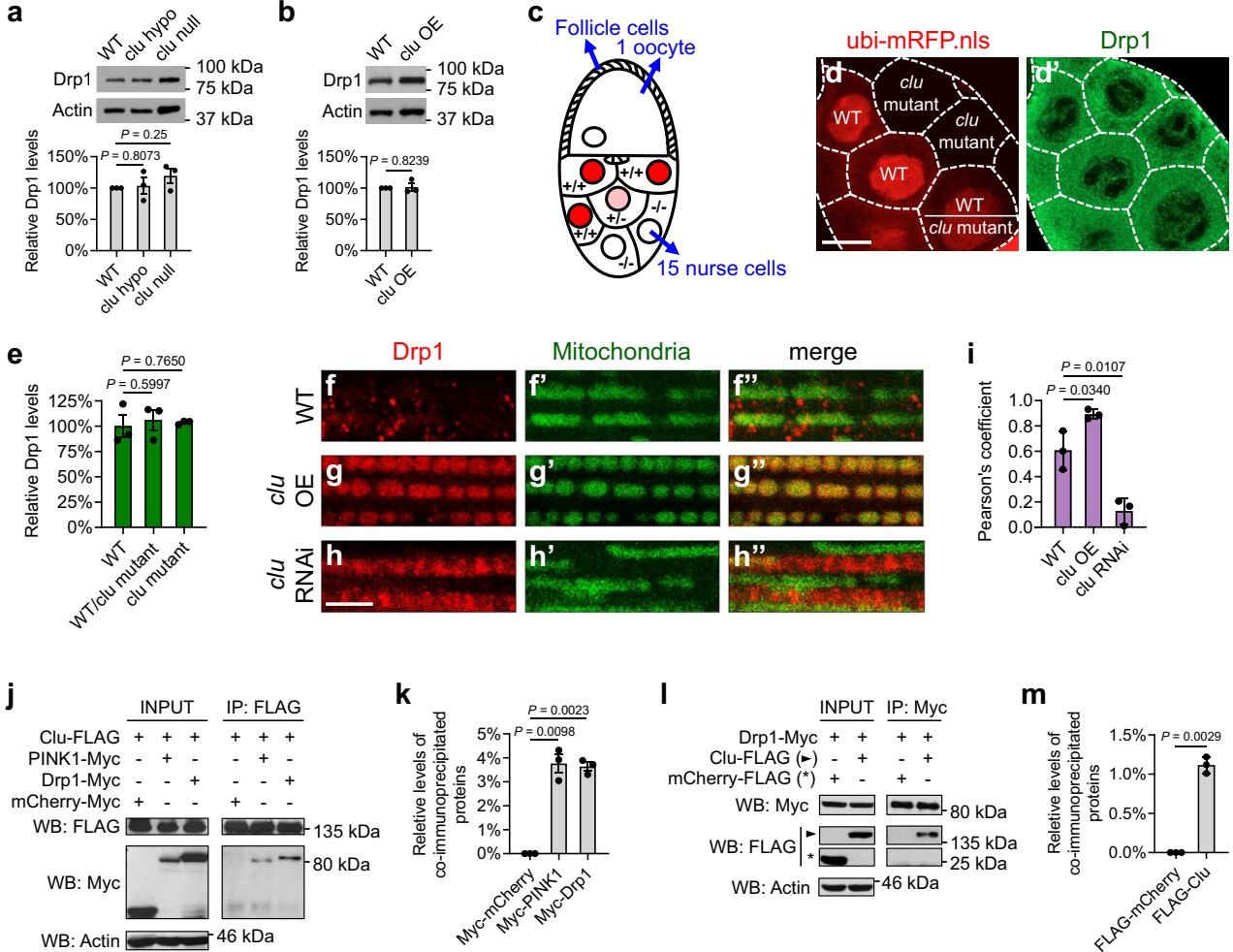

**Fig. 4 Clu promotes recruitment of Drp1 to mitochondria in *Drosophila*. a, b** Western blots and quantification of Drp1 levels in vivo, using lysates from fly thoraces expressing a FLAG-HA-tagged *drp1* genomic transgene in wild-type, *clu^f04554* null or *clu^d00713* hypomorphic mutant background (**a**), or with *clu* overexpression (**b**). Western blots were probed with mouse anti-HA and rabbit anti-actin antibodies. Images were quantified using Fiji/ImageJ, and Drp1 levels were normalized to actin levels (mean ± SEM, *n* = 3 independent experiments). **c** A schematic illustration of a stage 10a egg chamber, which displays a genetic mosaic performed with the FLP/FRT system. **d–d′** Confocal microscopy images of the same egg chamber labeled with *ubi-mRFP.nls* that indicates the genotypes of individual cells (red, **d**) and a mouse anti-FLAG antibody that indicates Drp1 levels (green, **d′**). White dashed lines illustrate the borders of individual nurse cells. Scale bar: 20 μM. **e** Drp1 immunofluorescence signals were quantified in three different egg chambers with a genetic mosaic using Fiji/ImageJ (mean ± SEM, *n* = 3). **f–h″** Confocal microscopy images of the thoracic muscle expressing the FLAG-HA-tagged *drp1* transgene. Drp1 is visualized using a mouse anti-HA antibody (red), and mitochondria are labeled with mitoGFP (green). Expression of UAS-mitoGFP, UAS-*clu*, and UAS-*clu* RNAi was driven by IFM-GAL4. Scale bar: 5 μM. **i** Pearson's coefficient indicates colocalization of Drp1 and mitochondria (mean ± SEM, *n* = 3 independent experiments as described in **f–h″**). In each independent experiment, 30–50 individual muscle pieces from a male fly were analyzed for each genotype, and the average Pearson's coefficient was calculated using Fiji/ImageJ. **j–m** Co-immunoprecipitation using lysates from *Drosophila* S2 cells transfected with the indicated plasmids. The INPUT represents 3% of each total lysate. FLAG-tagged Clu (**j**) and Myc-tagged Drp1 (**l**) co-IP each other with a mouse anti-FLAG antibody and a mouse anti-Myc antibody, respectively. Western blots were probed with rabbit anti-FLAG and rabbit anti-Myc antibodies. Experiments were performed in triplicate, and representative images are shown. **k, m** The levels of co-immunoprecipitated proteins were normalized to INPUT levels (mean ± SEM, *n* = 3). **a, e, i, k** One-way ANOVA with post hoc Tukey's HSD test. **b, m** Two-sided Student's *t*-test.

an early stage of oogenesis. Therefore, we performed a mosaic analysis in the female germline using the FLP/FRT system, such that all nurse cells can successfully complete oogenesis in the *clu* heterozygous background. Furthermore, *clu* null mutant cells and their neighboring *clu*-positive cells (an excellent internal control) can be labeled in an unambiguous fashion. As shown in Fig. 4c, in an egg chamber with genetic mosaic, a monolayer of somatic follicle cells coats the 16-cell germline cyst: 15 nurse cells are linked to a single oocyte through intracellular channels. Cells with the presence of the *ubi-mRFP.nls* transgene, visualized as red nuclei, contain one or two *clu^WT* (wild-type) alleles ("+/+" or "+/−"), while cells absent of red nuclei contain the *clu^f04554* null

mutant alleles ("−/−"). As shown in Fig. 4d–d', we observed no significant differences in Drp1 protein levels among these cells (quantified in Fig. 4e).

To determine whether *clu* regulates Drp1 translocation rather than protein levels, we further examined Drp1 localization in the muscle. While Drp1 is largely cytosolic under resting conditions, in response to signals that trigger mitochondrial fission, Drp1 is recruited onto mitochondria[31,32]. In wild-type flies, Drp1 showed punctate signals largely in the cytosol and occasionally on the OMM (Fig. 4f–f″). In contrast, *clu* OE resulted in Drp1 being preferentially localized onto the OMM (Fig. 4g–g″), and this was associated with a decrease in mitochondria size (compare Fig. 4g'

to Fig. 4f'). Conversely, *clu* knockdown resulted in a decrease in the levels of mitochondria-localized Drp1 (Fig. 4h–h", quantified in 4i). Consequentially, these muscle cells had more elongated mitochondria (compare Fig. 4h' to Fig. 4f').

Next, we examined whether Clu and Drp1 physically interact. As shown in Fig. 4j–m, Drp1 co-immunoprecipitated with Clu from S2 cell lysates and vice versa. PINK1 was used as a positive control and also co-immunoprecipitated with Clu[47]. The mCherry fusion proteins were used as negative controls and did not co-immunoprecipitate with Clu or Drp1, indicating that the protein interactions between Clu and Drp1 are specific rather than due to the tags used. Together, these results suggest that Clu and Drp1 form a protein complex, with Clu regulating mitochondrial recruitment of Drp1 but not changing Drp1 levels.

A previous study showed that *clu* negatively regulates Mfn (Marf) levels in *Drosophila*. This was argued to involve Clu binding to Valosin-containing protein (VCP), with VCP then promoting Mfn degradation[48]. VCP is a conserved ATPase mediating degradation of Mfn proteins during mitophagy[50] and *Drosophila* development[51]. However, we did not observe significant changes in Mfn levels in response to *clu* complete or partial loss-of-function in the muscle (Supplementary Fig. 2a). We also did not detect protein interactions between Clu and VCP in S2 cells by co-immunoprecipitation (co-IP), even though both proteins were abundantly overexpressed (Supplementary Fig. 2b). Therefore, we conclude that Clu regulates Drp1 rather than Mfn.

**CLUH regulates mitochondrial fission in mammalian cells.** To determine whether the function of *clu* in regulating mitochondrial fission is evolutionarily conserved, we examined mitochondrial morphology in response to *CLUH* loss-of-function or *CLUH* OE in human HeLa cells. We generated two independent *CLUH* knockout (KO) HeLa cell lines using the CRISPR–Cas9 system[52,53]. *CLUH-40* contains a single nucleotide deletion and *CLUH-43* contains a 4-nucleotide deletion, both of which are frameshift mutations leading to premature termination of translation (Fig. 5a). We confirmed the absence of CLUH expression by western blotting (Fig. 5b) and immunostaining (Supplementary Figure 3a–a'). These two *CLUH* KO HeLa cell lines showed similar phenotypes, including slow growth rate (data not shown) and clustered mitochondria (Fig. 5d; Supplementary Fig. 3e). We focused on *CLUH-43* (referred to as "*CLUH* KO cells" hereafter) for further studies.

Wild-type HeLa cells showed dispersed mitochondria throughout the cytosol (Fig. 5c), whereas *CLUH* KO HeLa cells exhibited severely clustered mitochondria in the perinuclear region, with a few long and interconnected mitochondria projecting towards the cell periphery (Fig. 5d). The mitochondrial phenotype in *CLUH* KO cells (Supplementary Fig. 3e–e') was completely suppressed by transfection of a plasmid expressing *CLUH* (Supplementary Fig. 3f–f'), confirming that the phenotype was indeed due to lack of *CLUH*. Intriguingly, the clustered mitochondria phenotype in *CLUH* KO cells was similar to that observed in HeLa cells overexpressing Drp1[K38A] (Fig. 5e), a dominant-negative form of Drp1 with defective GTP binding and impaired mitochondrial recruitment[54]. As a control, HeLa cells overexpressing Drp1[WT] did not show significant changes in mitochondrial morphology (Fig. 5f). The clustered mitochondria phenotype associated with Drp1[K38A] OE was also observed previously in COS-7 cells[4,55]. The phenotypic similarity between *CLUH* KO and Drp1[K38A] OE strongly suggests that in mammalian cells, the functions of CLUH and Drp1 are closely related.

The high density of mitochondria in the regions of clustered mitochondria (Fig. 5d, e) makes it difficult to examine mitochondrial morphology with high resolution. To overcome

this issue, we incubated cells with nocodazole, a tubulin-binding agent known to disperse mitochondrial clusters by depolymerizing microtubules[4,55]. Nocodazole treatment dispersed mitochondrial clusters in COS-7 cells[4] and HeLa cells (Fig. 5e'–e") overexpressing Drp1[K38A]. Similarly, *CLUH* KO HeLa cells exhibited elongated mitochondria after nocodazole treatment (Fig. 5d'–d"). To rigorously measure mitochondrial morphology changes, we used a morphology scoring assay to categorize cells as having punctate or fragmented (blue bar), tubular (red bar), elongated (green bar), or highly interconnected or clustered (purple bar) mitochondria (Fig. 5g). Indeed, the scoring assay showed that with nocodazole treatment, the majority of cells with *CLUH* KO or Drp1[K38A] OE had elongated mitochondria; while in control cells, nocodazole treatment did not alter mitochondrial morphology significantly (Fig. 5c'–c", f–f").

To further support the above observation, we examined mitochondrial morphology due to a partial loss-of-function of *CLUH*. Cells with control siRNA mostly showed tubular mitochondria (Fig. 5h–h'). In contrast, *CLUH* siRNA resulted in a significant increase in the proportion of cells having elongated or interconnected mitochondria, without mitochondrial clustering (Fig. 5i–i', quantified in k).

We then examined mitochondrial phenotypes in response to *CLUH* OE. *CLUH* OE in HeLa cells resulted in mitochondria with a significant reduction in size (Fig. 5j–j'; Supplementary Fig. 3d), as well as a significant increase in the proportion of cells with punctate or fragmented mitochondria (Fig. 5k). *CLUH* OE and *CLUH* knockdown were confirmed using western blotting (Supplementary Fig. 3b) and immunostaining (Supplementary Fig. 3c'–f', g–j). Taken together, these results show that *CLUH* regulates mitochondrial morphology in a pro-fission manner.

**Overexpression of Drp1[S637A] suppresses the mitochondrial clustering phenotype in CLUH KO cells.** Next, we asked whether *CLUH* and *drp1* function in the same pathway in mammalian cells, similar to our observations of the *clu-drp1* genetic interactions in *Drosophila*. Accordingly, we tested if the OE of Drp1 can rescue the clustered mitochondria in *CLUH* KO cells. Because OE of Drp1[WT] did not result in significantly altered mitochondrial morphology in HeLa cells (Fig. 5f), we instead overexpressed the phospho-dead Drp1[S637A] mutant in *CLUH* KO cells. In mammalian cells, the localization of Drp1 is tightly controlled by its phosphorylation state, with dephosphorylation of Drp1 at serine 637 resulting in constant mitochondrial localization of Drp1 and mitochondrial fragmentation[56]. Indeed, the OE of Drp1[S637A] significantly rescued the clustered mitochondria in *CLUH* KO cells (Fig. 5n, compared to Fig. 5m), but OE of Drp1[WT] did not (Fig. 5o, quantified in p). This suggests that *CLUH* and *drp1* function in the same pathway in mammalian cells, with *drp1* being a downstream target of *CLUH*.

**CLUH complexes with Drp1 and promotes recruitment of Drp1 to mitochondria in mammalian cells.** Next, we investigated if *CLUH* regulates mitochondrial morphology through Drp1 in mammalian cells, as *clu* does in *Drosophila*. We first determined whether *CLUH* regulates the expression levels of Drp1. No significant changes of Drp1 levels were observed in HeLa cells in response to either *CLUH* loss-of-function or *CLUH* OE (Fig. 6a, b). *CLUH* also did not regulate levels of Mfn1 or Mfn2 (homologs in mammalian cells) (Supplementary Fig. 4a, b). Co-IP experiments in HeLa cells showed that CLUH bound Drp1 (directly or indirectly) (Fig. 6c, d) but not Mfn2 (Supplementary Fig. 4c), indicating that CLUH and Drp1 form a protein complex in mammalian cells, as with Clu and Drp1 in *Drosophila*.

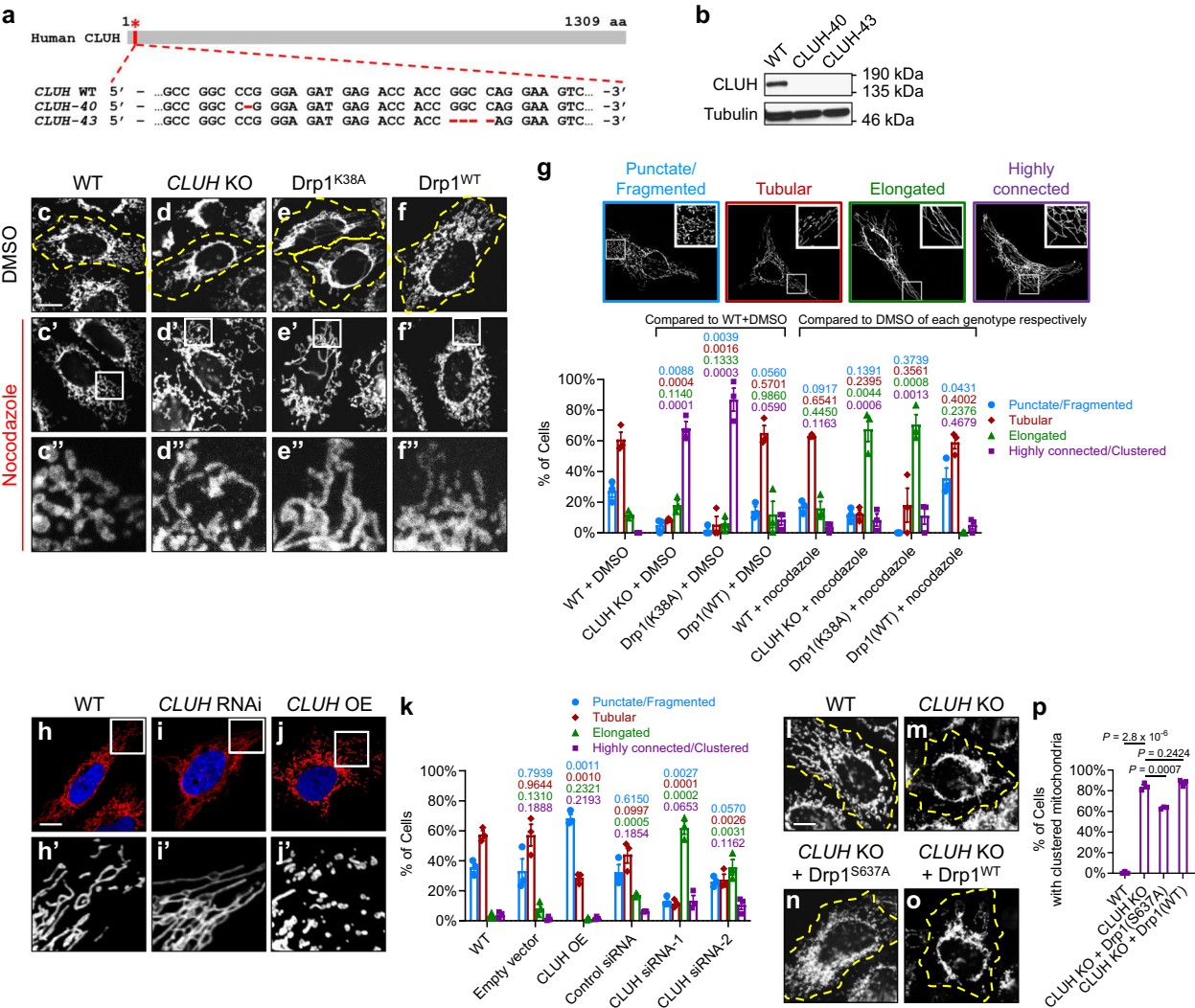

**Fig. 5 *CLUH* regulates mitochondrial morphology in mammalian cells. a** Generation of two *CLUH* knockout (KO) HeLa cell lines, *CLUH-40* and *CLUH-43*, using the CRISPR–Cas9 system[52,53]. Positions of the deletions in the CLUH protein (red asterisk) and in the *CLUH* gene (within the second exon, red dashed lines) are shown. **b** Western blots using a rabbit anti-CLUH antibody confirm the absence of CLUH in both *CLUH* KO cell lines. Experiments were repeated independently 5 times, and representative images are shown. **c–f** Confocal microscopy images showing mitochondrial morphology in HeLa cells of the indicated genotypes under resting conditions. Yellow dashed lines mark the boundaries of cells. **c'–f'** Mitochondrial morphology after nocodazole treatment. **c''–f''** Enlarged views of the boxed regions in (**c'–f'**). **c–f''** Mitochondria are labeled with a mouse anti-TOM20 antibody. **g** Mitochondrial morphology in (**c–f'**) is quantified using the mitochondrial morphology scoring assay, and one representative image for each category of cells is shown above the bar graph. **h–j'** Confocal microscopy images showing mitochondrial morphology in response to *CLUH* RNAi or *CLUH* overexpression. Mitochondria are labeled by transfection of a plasmid expressing Mito-DsRed (red), and the nuclei are labeled with DAPI (blue). **h'–j'** Enlarged views of the boxed regions in (**h–j**). **k** Mitochondrial morphology in (**h–j'**) is quantified using the scoring assay. **l–o** Expression of Drp1[S637A] (**n**), but not Drp1[WT] (**o**), rescues mitochondrial clustering in *CLUH* KO HeLa cells. Mitochondria are labeled with a mouse anti-TOM20 antibody. Yellow dashed lines mark the boundaries of cells. **p** Quantification of the percentage of cells showing clustered mitochondria for each indicated genotype. **c–f''**, **h–j'**, **l–o** Experiments were performed in triplicate, and representative images are shown. Scale bars: 10 µM. **g**, **k**, **p** In the mitochondrial morphology scoring assay, numbers of cells in each of the four categories were counted, with a total number of >200 cells in each experiment (mean ± SEM, $n = 3$ independent experiments). Statistical analysis was carried out using one-way ANOVA with post hoc Tukey's HSD test, with *P*-values displayed in the graphs. $P < 0.05$: significantly different from the same morphology category in the control group.

We next determined if *CLUH* regulates mitochondrial recruitment of Drp1, by measuring levels of mitochondrially bound Drp1 in wild-type (Fig. 6e–f'') and *CLUH* KO (Fig. 6g–h''') HeLa cells treated with digitonin. Digitonin is a detergent that permeabilizes the plasma membrane, such that cytosolic Drp1 can be eliminated through subsequent washes and only OMM-bound Drp1 remains in the cells for enhanced visualization and precise measurement[31,57]. By quantifying the immunofluorescence intensity of the retained Drp1, we found that OMM-bound

Drp1 was significantly reduced in *CLUH* KO cells (compare Fig. 6g''', h–h''' to Fig. 6e''', f–f'''; quantified in Fig. 6i, j). These results indicate that *CLUH* is required for mitochondrial recruitment of Drp1.

We then asked whether *CLUH* OE increases Drp1 recruitment to the OMM. In wild-type HeLa cells, punctate Drp1 staining was observed mostly in the cytoplasm, with the occasional signals on the OMM (Fig. 6k–k'', l–l''). In contrast, in *CLUH*-overexpressing cells, Drp1 showed significantly increased co-localization with

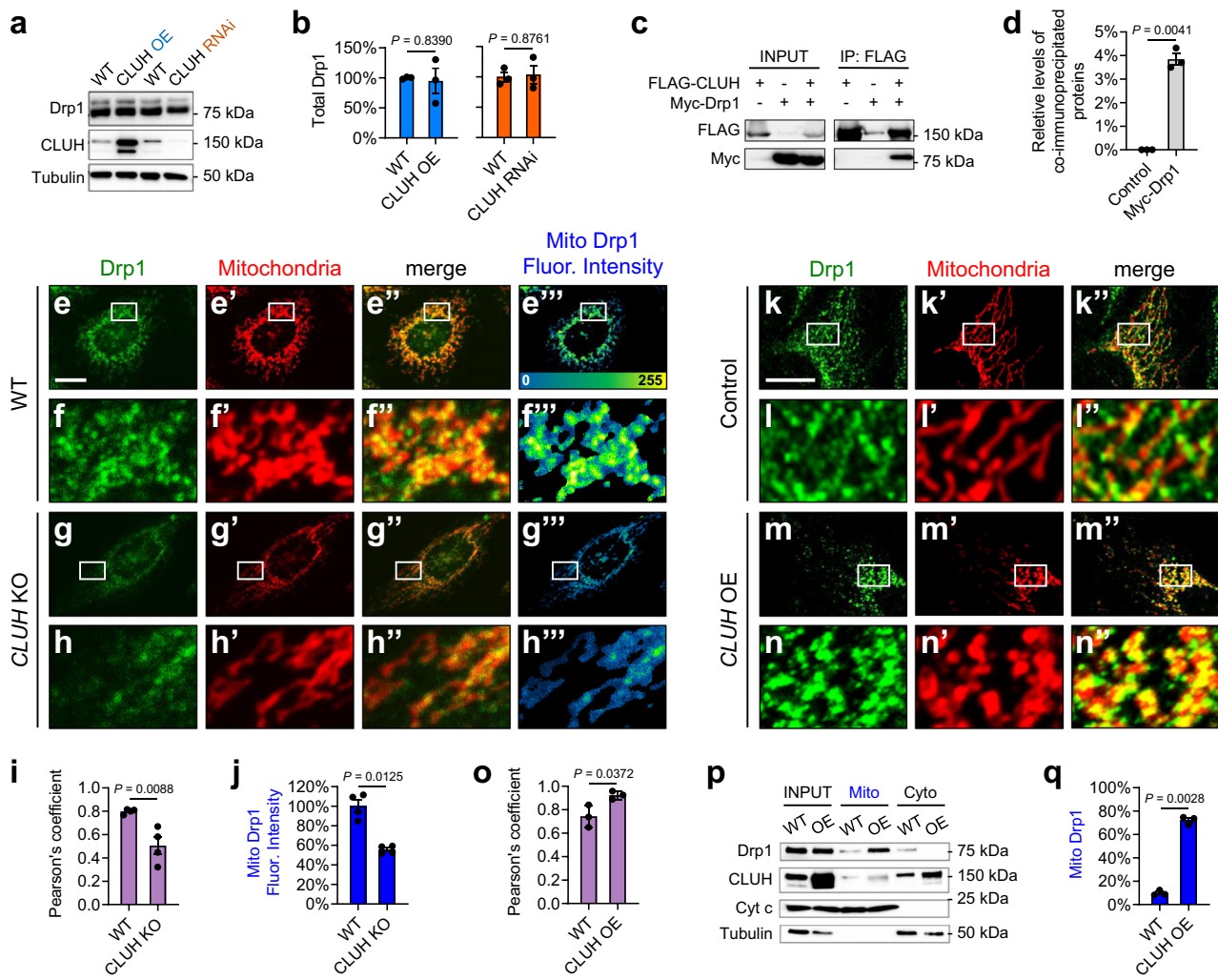

**Fig. 6 CLUH promotes the recruitment of Drp1 to mitochondria in mammalian cells. a**, **b** Western blots showing Drp1 protein levels in response to *CLUH* overexpression or *CLUH* RNAi. Drp1 levels were normalized to Tubulin levels (mean ± SEM, n = 3 independent experiments). **c**, **d** Co-immunoprecipitation using lysates from HeLa cells transfected with the indicated constructs. The INPUT represents 5% of each total lysate. Myc-Drp1 was co-immunoprecipitated with FLAG-CLUH by using a mouse anti-FLAG antibody. The levels of co-immunoprecipitated proteins were normalized to INPUT levels (mean ± SEM, n = 3 independent experiments). **e–h‴** Confocal microscopy images showing levels of mitochondrially bound Drp1 ("Mito Drp1") in HeLa cells with digitonin treatment. Drp1 and mitochondria are labeled using a mouse anti-Drp1 antibody (green) and a goat anti-Hsp60 antibody (red), respectively. **f–f‴**, **h–h‴** Enlarged views of the boxed regions in (**e–e‴**, **g–g‴**). **e‴–h‴** The heat map reflects the fluorescence intensity of "Mito Drp1". **i** Pearson's coefficient indicates colocalization of Drp1 and mitochondria. **j** The fluorescence intensity in the heat map (**e‴**, **g‴**) was quantified to measure "Mito Drp1" levels. **i**, **j** Fiji/ImageJ was used for image analysis (mean ± SEM, n = 4 independent experiments). In each experiment, 12–25 cells were measured for each genotype, and the average Pearson's coefficient (**i**) or fluorescence intensity (**j**) was calculated. **k–n″** Confocal microscopy images showing Drp1 localization. Drp1 and mitochondria are labeled using a mouse anti-Drp1 antibody (green) and by transfection of a plasmid expressing Mito-DsRed (red), respectively. **i–i″**, **n–n″** Enlarged views of the boxed regions in (**k–k″**, **m–m″**). **o** Pearson's coefficient indicates colocalization of Drp1 and mitochondria (mean ± SEM, n = 3 independent experiments). **p**, **q** Fractionation of HeLa cell lysates, showing total Drp1 levels (INPUT) and levels of Drp1 in the mitochondrial (Mito) and the cytoplasmic (Cyto) fractions. Western blots were probed with mouse anti-Drp1 and rabbit anti-CLUH antibodies. A mitochondrial protein, Cytochrome c (Cyt c), and a cytoplasmic protein, α-Tubulin, serve as controls. Drp1 levels in the Mito fraction were normalized to total Drp1 levels (mean ± SEM, n = 3 independent experiments). (**e–h‴**, **k–n″**) Scale bars: 10 μM. **b**, **d**, **i**, **j**, **o**, **q** Two-sided Student's *t*-test.

mitochondria (Fig. 6m–m", n–n"), indicating increased recruitment of Drp1 onto mitochondria in these cells, as compared to that in the wild-type cells (Fig. 6o). Accordingly, mitochondria are smaller in size in response to *CLUH* OE (compare n' with i'). Consistently, in a cell fractionation assay, we found that *CLUH* OE resulted in significantly increased Drp1 levels in the mitochondrial fraction and correspondingly decreased Drp1 levels in the cytosolic fraction (Fig. 6p, quantified in q). Total Drp1 levels were comparable in wild-type and *CLUH*-overexpressing cells (Fig. 6p; also Fig. 6a, b). Together, these results

demonstrate that CLUH promotes the recruitment of Drp1 to the OMM to regulate mitochondrial fission, and this mechanism is conserved from *Drosophila* to mammalian cells.

**Drp1 receptors MiD49 and Mff mediate interactions between CLUH and Drp1.** Drp1 is localized predominantly in the cytosol and is recruited onto mitochondria via interactions with Drp1 receptors anchored to the OMM. Modulation of Drp1 receptors is key to controlling the recruitment of Drp1 from the cytosol onto

the OMM[32]. Therefore, to explore the molecular mechanisms by which CLUH regulates mitochondrial recruitment of Drp1, we went on to test whether CLUH interacts with Drp1 receptor(s). If CLUH indeed binds Drp1 receptor(s), these protein interactions may facilitate mitochondrial recruitment of Drp1.

In mammalian cells, Drp1 receptors include Mff, MiD49, and MiD51[28–32], which all localize to the OMM. *Drosophila* has Mff[28,58], but not MiD49 or MiD51. As the role of *Drosophila* Mff in recruiting Drp1 remains unclear[28], we focused our studies on interactions between CLUH and Drp1 receptors using HeLa cells. We carried out co-IP experiments using lysates of HeLa cells expressing CLUH along with Mff, MiD49, or MiD51. We found that CLUH bound MiD49 (Fig. 7a, b), as well as Mff (Fig. 7c, d), but not MiD51 (Supplementary Fig. 5a). As shown in Fig. 6c, d, CLUH also bound Drp1. Together, these co-IP data suggest that CLUH and Drp1 form a protein complex with MiD49 and/or Mff.

To further investigate these protein interactions, we carried out in vitro protein binding assays in a cell-free system using recombinant GST- or His- fusion proteins purified from *E. coli*. The transmembrane (TM) domains of MiD49 and Mff were deleted in order to yield more soluble proteins. Purified CLUH, Drp1, and MiD49 with a TM deletion (MiD49ΔTM) were included together in the in vitro system, with MiD49ΔTM as the bait for the pulldown. MiD49ΔTM pulled down comparable levels of CLUH and Drp1 simultaneously (Fig. 7e; Supplementary Fig. 5b). We next included purified CLUH and Drp1 together with Mff in the in vitro system. Detection of in vitro binding between Drp1 and Mff was hindered by the transient or unstable nature of the Drp1–Mff complex. Such physical interactions had only been previously detected when the complex was stabilized by cross-linking agents[28,29]. However, a recent study showed that deletions of the insert B domain from Drp1 (Drp1ΔIB) and of the coiled-coil domain from Mff (MffΔCC) enabled detection of robust Drp1–Mff binding in vitro, in the absence of cross-linkers[57]. Therefore, we used Drp1ΔIB, and Mff with either a TM deletion alone (MffΔTM) (Fig. 7f), or with both CC and TM deletions (MffΔCCΔTM) (Supplementary Figure 5c), in this assay. As with MiD49 (Fig. 7e; Supplementary Fig. 5b), Mff pulled down similar levels of CLUH and Drp1 simultaneously (Fig. 7f; Supplementary Fig. 5c). Because direct binding between Drp1 and its receptors has been demonstrated using in vitro pulldowns[57] and cryo-electron microscopy structures[32], our results in Fig. 7e, f suggest two possible scenarios. One is that MiD49 and Mff directly bind both CLUH and Drp1, thus pulling down both proteins at the same time. Alternatively, MiD49 and Mff pull down CLUH indirectly, with Drp1 serving as a bridging molecule. To distinguish between these possibilities, we included CLUH together with MiD49 or Mff in the in vitro system, without Drp1. Mff and MiD49 each pulled down CLUH (Supplementary Fig. 5d), indicating direct binding between CLUH and these receptors. However, when CLUH and Drp1 were included together in vitro, Drp1 was not able to pull down CLUH in this system (Fig. 7g), suggesting that the physical interactions between CLUH and Drp1 are indirect. Together, these results show that Mff and MiD49 directly bind both CLUH and Drp1, which possibly bridge CLUH–Drp1 interactions and bring about the recruitment of Drp1 to the OMM.

We further tested this model of CLUH–Mff/MiD49–Drp1 protein interactions via proximity ligation assays (PLA) in HeLa cells (Fig. 7h–h'). PLA allows for the detection of endogenous protein interactions with single-molecule resolution through a 1000-fold amplification of fluorescence signals that occurs only if the two proteins of interest are in close proximity. When PLA was carried out with both CLUH and Drp1 antibodies, punctate signals were observed throughout the cytoplasm, indicating the

proximity of CLUH and Drp1 (Fig. 7h). In contrast, no PLA signals were detected in the control experiments with either CLUH or Drp1 antibodies alone, indicating that the signals were specific to CLUH–Drp1 interactions. We did not observe significant co-localization between CLUH–Drp1 PLA signals and mitochondria (Fig. 7h'), suggesting that these protein interactions occurred in the cytosol or near mitochondria, but not on the OMM. Importantly, knockdown of *MiD49* resulted in a significantly decreased frequency of protein interactions between CLUH and Drp1 (Fig. 7i–i', quantified in Fig. 7j). This observation substantiates the idea that interactions between CLUH and Drp1 are bridged by Drp1 receptors.

To determine, in an in vivo setting, if CLUH alone was sufficient to recruit Drp1, we targeted Mff, MiD49, or CLUH to lysosomes by fusion to lysosomal-associated membrane protein 1 (LAMP1) and tested whether Drp1 could be ectopically recruited to lysosomes in HeLa cells. This provides a robust assay to test the sufficiency of factors to recruit Drp1 based on direct protein interactions[57]. LAMP1–Mff (Fig. 7k–k") and LAMP1–MiD49 (Fig. 7l–l") each successfully recruited Drp1 to lysosomes[57]. However, LAMP1–CLUH alone was not able to recruit Drp1 (Fig. 7m–m"). As expected, we observed significantly elongated, interconnected, and/or clustered mitochondria in HeLa cells with expression of LAMP1–Mff (Supplementary Fig. 6b–b", compared to a–a") or LAMP1–MiD49 (Supplementary Fig. 6c–c"). These phenotypes reflect a significant portion of Drp1 being sequestered on lysosomes, with a correspondingly decreased amount of Drp1 being recruited to mitochondria. In contrast, expression of LAMP1–CLUH did not result in changes of mitochondrial morphology as compared to non-transfected cells (Supplementary Fig. 6d–d", quantified in e), because LAMP1–CLUH did not sequester Drp1 on lysosomes. These results further support the idea that unlike the direct interactions between Mff/MiD49 and Drp1, interactions between CLUH and Drp1 are indirect. CLUH does not serve as a receptor to recruit Drp1 directly, but instead, CLUH regulates Drp1 translocation through bridging molecules such as Mff and MiD49.

**CLUH regulates MiD49 and Mff protein levels in mammalian cells.** We next investigated the mechanism by which MiD49 and Mff mediate mitochondrial recruitment of Drp1 by CLUH. Levels of Drp1 receptors directly determine the amount of Drp1 being recruited onto mitochondria, as shown by previous work in which decreased protein levels of Drp1 receptors lead to reduced mitochondrial recruitment of Drp1, and increased levels of Drp1 receptors result in more Drp1 recruitment onto mitochondria[31]. Therefore, we measured protein levels of Mff and MiD49 in HeLa cells in response to changes in *CLUH* levels. Immunofluorescence results showed that *CLUH* KO led to a dramatic decrease in MiD49 protein levels (Fig. 8a, b), and a milder but significant decrease in Mff protein levels (Fig. 8c, d, quantified in e). Western blotting results also showed that *CLUH* KO led to significantly decreased protein levels of Mff and MiD49. Conversely, *CLUH* OE resulted in significantly increased MiD49 levels (Fig. 8f, g). Therefore, CLUH positively regulates Mff and MiD49 protein levels.

To test whether Mff and MiD49 are subject to transcriptional regulation by CLUH, we performed quantitative PCR (qPCR) but observed no significant changes of *Mff* or *MiD49* mRNA levels in response to either *CLUH* OE or *CLUH* KO (Fig. 8h). These results indicate that CLUH regulates Mff and MiD49 protein levels through posttranscriptional mechanisms, which could involve regulation of translation, mRNA localization, or protein degradation. CLUH has been shown to bind nuclear-encoded, mitochondria-targeted mRNAs and regulate their localized

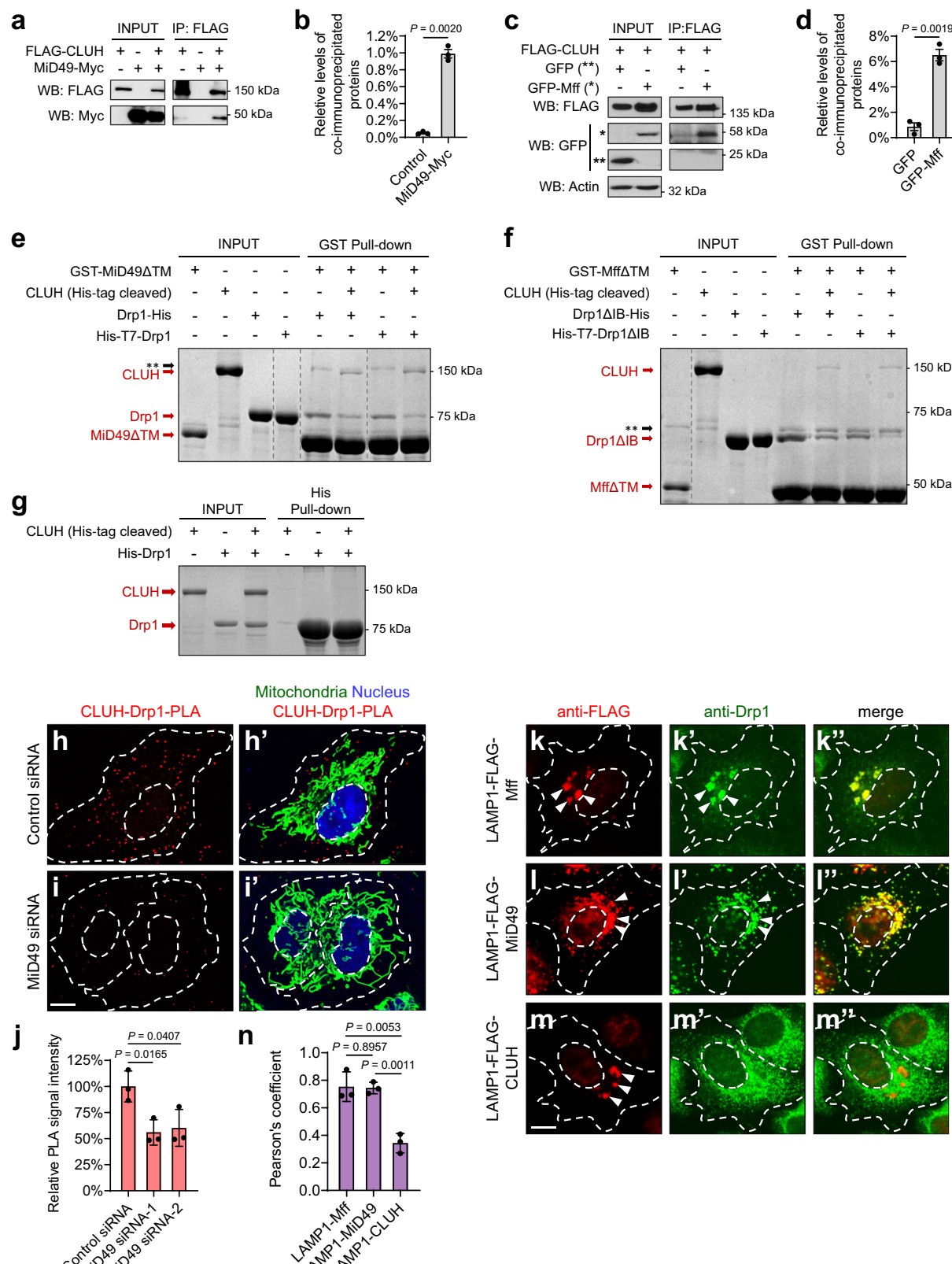

translation near mitochondria[39,43,45]. To examine if *Mff* and *MiD49* mRNAs are targets bound by CLUH, we performed RNA immunoprecipitation (RIP). With RIP, the mRNA targets of an RNA-binding protein can be captured via immunoprecipitation of the protein of interest. Indeed, we found that CLUH bound *Mff* and *MiD49* mRNAs (Fig. 8i, j). Next, to determine if CLUH

regulates translation of Mff and MiD49, we carried out puromycylation-PLA (puro-PLA). When HeLa cells are treated with puromycin (a tRNA analog), newly synthesized proteins are puromycylated and can be recognized by an anti-puromycin antibody. When the anti-puromycin antibody is combined with either an anti-Mff or an anti-MiD49 antibody in the PLA assay,

**Fig. 7 CLUH directly binds MiD49 and Mff, which mediates the interactions between CLUH and Drp1. a–d** Co-immunoprecipitation using lysates from HeLa cells transfected with the indicated constructs. The INPUT represents 5% (**a**) and 3% (**c**) of each total lysate, respectively. Both MiD49-Myc (**a**, **b**) and GFP-Mff (**c**, **d**) were co-immunoprecipitated with FLAG-CLUH by using a mouse anti-FLAG antibody. Western blots were probed with rabbit anti-FLAG, Myc, GFP, and actin antibodies. The levels of co-immunoprecipitated proteins were normalized to INPUT levels (mean ± SEM, $n = 3$ independent experiments, two-sided Student's $t$-test). **e** In a cell-free system with recombinant GST- or His-fusions purified from *E. coli*, MiD49 pulls down both Drp1 and CLUH in vitro. **f** With purified Mff (isoform-e), CLUH, and Drp1 as INPUT, Mff pulls down both Drp1 and CLUH in vitro. **e**, **f** Gray dashed lines delineate the boundaries between non-adjacent lanes in the same gel. The whole gel images are shown in Supplementary Fig. 5b, c. Double asterisks indicate the position of a non-specific band. IB denotes Insert B domain. TM denotes transmembrane domain. All the above experiments were performed without the use of cross-linkers. **g** Drp1 does not pull down CLUH in vitro. **e–g** Experiments were repeated independently 3 times, and representative images are shown. **h–i'** CLUH-Drp1-PLA (red) was carried out using rabbit anti-CLUH and mouse anti-Drp1 antibodies in HeLa cells transfected with control or *MiD49* siRNA. Mitochondria: goat anti-Hsp60 antibody (green). Nucleus: DAPI (blue). **j** Quantification of CLUH-Drp1-PLA signals. **k–m"** FLAG-tagged Mff, MiD49 and CLUH fused to LAMP1 are visualized with a rabbit anti-FLAG antibody (red), and Drp1 is visualized with a mouse anti-Drp1 antibody (green). White arrowheads indicate lysosome-localized proteins. **n** Pearson's coefficient indicates colocalization of the indicated LAMP1 fusion proteins and Drp1. **h–i'**, **k–m"** White dashed lines mark the boundaries of the cells and the nuclei. Scale bars: 10 µM. **j**, **n** Fiji/ImageJ was used for image analysis (mean ± SEM, $n = 3$ independent experiments, one-way ANOVA with post hoc Tukey's HSD test). In each experiment, 20–30 cells were analyzed for each genotype and the average PLA signal intensities (**j**) or Pearson's coefficient (**n**) was calculated.

only newly synthesized Mff or MiD49 proteins are highlighted in situ[59]. Our puro-PLA results showed significantly reduced puromycylation of Mff (Fig. 8k–l') and MiD49 (Fig. 8m–n') in *CLUH* KO cells as compared with control cells, indicating that translation of Mff and MiD49 was significantly decreased in response to *CLUH* KO (Fig. 8o). Importantly, Mff and MiD49 translation occurred in the cytosol or near mitochondria but did not exactly co-localize with mitochondria (Fig. 8k', l', m', n', quantified in p).

Next, we investigated if CLUH regulates *Mff* and *MiD49* mRNA localization using single-molecular fluorescence in situ hybridization (FISH) (RNAscope). Consistent with our qPCR results, the number of detected *Mff* and *MiD49* mRNA molecules did not show significant differences in cells with *CLUH* KO or OE, as compared to control HeLa cells (Supplementary Figure 7a–c, d–f). We also examined the distribution of *Mff* and *MiD49* mRNAs in relation to mitochondria but did not observe significant alterations in response to either *CLUH* KO or *CLUH* OE (Supplementary Fig. 7a'–c', d'–f', quantified in g).

Based on these multiple lines of experimental evidence, we conclude that CLUH binds *Mff* and *MiD49* mRNAs, and regulates their protein levels through mechanisms involving translational control. The TPR domain of CLUH has been shown to be essential for RNA binding. Indeed, we found that, unlike full-length CLUH, OE of CLUHΔTPR did not result in mitochondrial fragmentation in HeLa cells (Supplementary Fig. 7h, i). This result further demonstrates that the RNA-binding activity of CLUH is essential for the regulation of Drp1 receptors, Drp1 translocation, and therefore mitochondrial fission. Our results strongly argue for translational control of Mff and MiD49 by CLUH; however, other contributing mechanisms, such as regulation of protein degradation, cannot be completely ruled out.

## Discussion

Drp1 is the master regulator driving mitochondrial fission, which in turn controls mitochondrial morphology and quality, metabolic homeostasis, and organismal health. In this study, we identified Clu/CLUH as a key evolutionarily conserved regulator of Drp1. This conclusion is supported by the following key findings: (1) In *Drosophila*, adult lethality, severe muscle degeneration, and mitochondrial defects of *clu* null mutants are significantly suppressed by *drp1* OE, indicating a crucially important role for Clu in regulating Drp1 to maintain mitochondrial integrity and tissue health. (2) As with loss of *drp1*, loss of *clu/CLUH* results in elongated mitochondria, while as with *drp1* OE, *clu/CLUH* OE leads to fragmented mitochondria in *Drosophila*

and mammalian cells, indicating a role of *clu/CLUH* in regulating mitochondrial morphology. (3) Mechanistically, Clu/CLUH regulates the recruitment of Drp1 onto mitochondria but not Drp1 expression levels. (4) Furthermore, CLUH regulates mitochondrial recruitment of Drp1 by direct binding to both mRNA and protein for Drp1 receptors Mff and MiD49, and by regulating their expression levels through translational control. (5) Last but not least, as with *drp1* OE, *clu* OE rescues mitochondrial and tissue defects of both *PINK1* and *parkin* null mutants in *Drosophila*, highlighting the potential therapeutic value of manipulating *CLUH* and *drp1* for the treatment of PD.

Recruitment of Drp1 from the cytosol to mitochondria plays a critical role in the regulation of mitochondrial fission and is controlled by multiple mechanisms, including post-translational modifications of Drp1[60–62], calcium signaling[60], ER–mitochondria interactions[63,64], and mitochondria–cytoskeleton interactions[15,16]. Among the above regulatory mechanisms, many ultimately involve modulation of interactions between Drp1 and its receptors, thereby controlling the recruitment of Drp1 onto the OMM from the cytosol[32]. Levels of Drp1 receptors determine the amount of Drp1 that is recruited onto the OMM[31]. However, factors controlling Drp1 receptor interactions and receptor levels remain unknown. Here, we identified CLUH as a new regulator that interacts with Drp1 receptors and modulates their levels, thus promoting the recruitment of Drp1 onto the OMM while not changing Drp1 levels. CLUH directly interacts with both mRNA and protein of Drp1 receptors and controls levels of these Drp1 receptors through translational regulation (Fig. 9a). Loss of *CLUH* results in defective translation of Mff and MiD49 (Fig. 8k–n'), leading to decreased Mff and MiD49 protein levels (Fig. 8a–g). This further results in reduced mitochondrial recruitment of Drp1 (Figs. 4 and 6), defective mitochondrial fission (Figs. 2 and 5), and tissue damage (Fig. 3 and Fig. 9b).

Mounting evidence suggests that Mff, MiD49, and MiD51 play nonredundant roles in their interactions with Drp1, with each receptor recruiting a subset of post-translationally modified Drp1[57], and with each receptor requiring different cofactors. For instance, MiD51, but not MiD49, requires ADP as a cofactor to activate the GTPase activity of Drp1[65]. The binding specificity of CLUH for Mff and MiD49, but not MiD51, may indicate that CLUH only facilitates mitochondrial recruitment of a subset of Drp1 which undergoes specific post-translational modifications or requires specific co-factors. Alternatively, CLUH may only recruit Drp1 under certain physiological conditions. Differential regulation of different Drp1 receptors may provide a mechanism for fine-tuning Drp1 recruitment and activity in response to different subcellular stimuli.

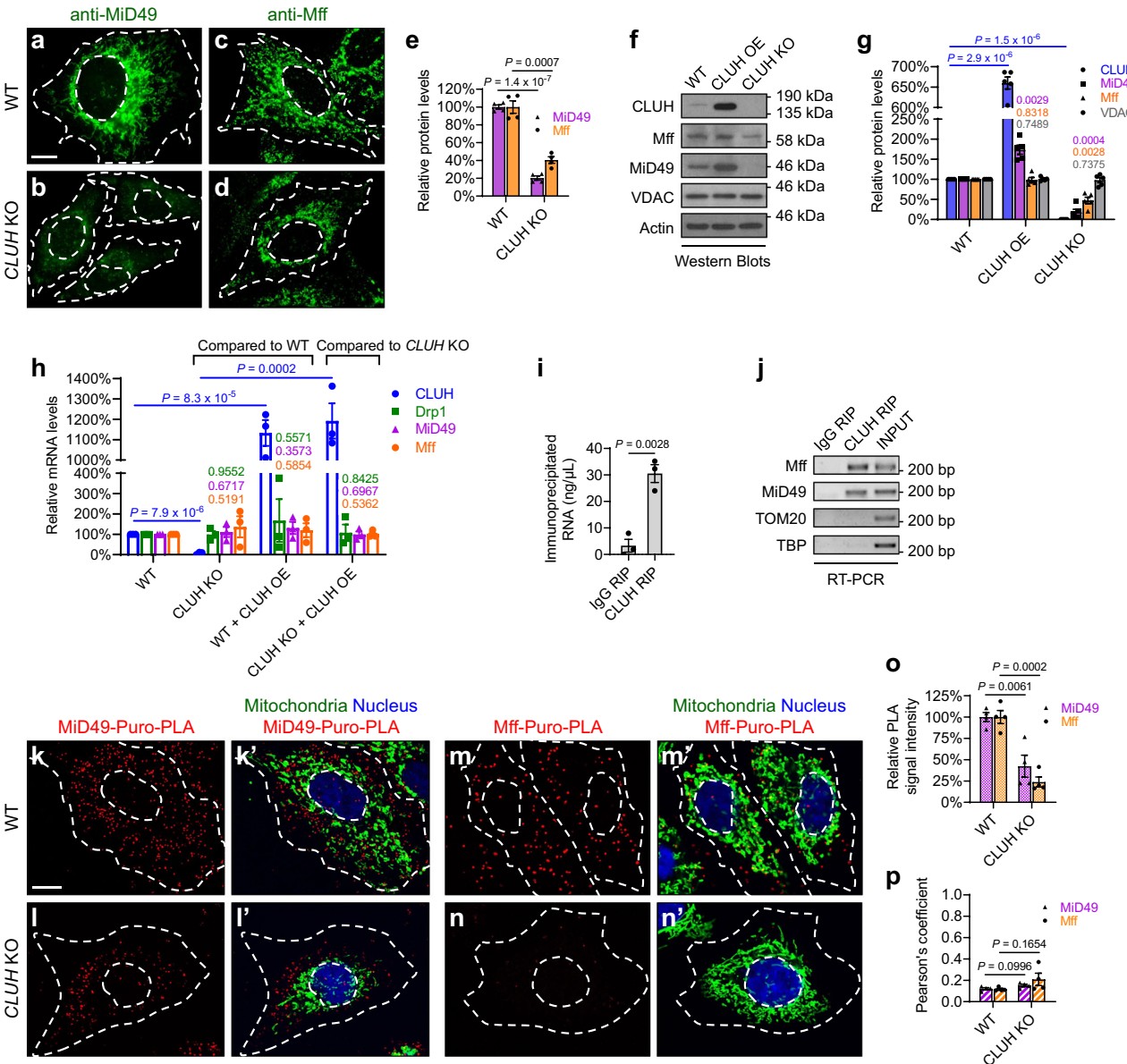

**Fig. 8 CLUH controls MiD49 and Mff protein levels through translational regulation in mammalian cells. a–d** Immunofluorescence (IF) staining showing endogenous MiD49 and Mff protein levels in wild-type and *CLUH* KO cells, using anti-MiD49 and anti-Mff antibodies, respectively. **e** Quantification of IF signals of MiD49 and Mff, normalized to IF signals of a control mitochondrial protein TOM20 (images not shown) in the same cell, using Fiji/ImageJ. **f, g** Western blots showing Mff and MiD49 protein levels in response to *CLUH* overexpression or *CLUH* KO. VDAC was used as a loading control for mitochondrial mass, and actin was used as a housekeeping protein loading control. Western blot experiments were repeated independently 5 times, with representative images shown in (**f**) and statistical analysis shown in (**g**) (mean ± SEM, $n = 5$). **h** Quantitative PCR (qPCR) experiments showing *Drp1*, *Mff*, and *MiD49* mRNA levels in response to *CLUH* overexpression or *CLUH* KO. Total RNAs extracted from HeLa cells of the indicated genotypes were used as templates for the synthesis of cDNAs, which were then used as templates for qPCR. Experiments were repeated in triplicate (mean ± SEM, $n = 3$). **i, j** RNA Immunoprecipitation (RIP) experiments show that CLUH binds mRNAs of *MiD49* and *Mff*, but not mRNAs of control genes *TOM20* or *TBP* (encoding the nuclear TATA-binding protein). Experiments were repeated in triplicate (mean ± SEM, $n = 3$), and representative images are shown in (**j**). **k–n′** Puro-PLA was carried out using a mouse anti-puromycin antibody in combination with either a rabbit anti-MiD49 antibody (**k–l′**) or a rabbit anti-Mff antibody (**m–n′**) (red). Mitochondria: goat anti-Hsp60 antibody (green). Nucleus: DAPI (blue). **o** Quantification of relative Puro-PLA signals using Fiji/ImageJ. **p** Pearson's coefficient indicates colocalization of Puro-PLA signals and mitochondria. **a–d, k–n′** Experiments were repeated independently for 4 times, and representative images are shown (mean ± SEM, $n = 4$). In each experiment, 20–30 cells were analyzed for each genotype, and the average IF signal intensity (**e, o**) or Pearson's coefficient (**p**) was calculated. Scale bars: 10 μM. **e, i, o, p** Two-sided Student's *t*-test. **g, h** One-way ANOVA with post hoc Tukey's HSD test, with *P*-values displayed in the graphs.

While MiD49 and Mff are both regulated by CLUH, we do observe some differences in terms of the strength of regulation. In HeLa cells, changes in Mff levels are more subtle than those of MiD49 in response to *CLUH* KO or OE, which are consistently seen in our immunofluorescence (Fig. 8a–e) and western blotting

(Fig. 8f, g) experiments. In addition, our RNAscope assays show that *Mff* mRNAs appear as small, discrete puncta (Supplementary Fig. 7d–f), whereas *MiD49* mRNAs are found as larger particles, where multiple mRNAs may cluster together (Supplementary Fig. 7a–c). CLUH has been shown to bind multiple mRNAs of

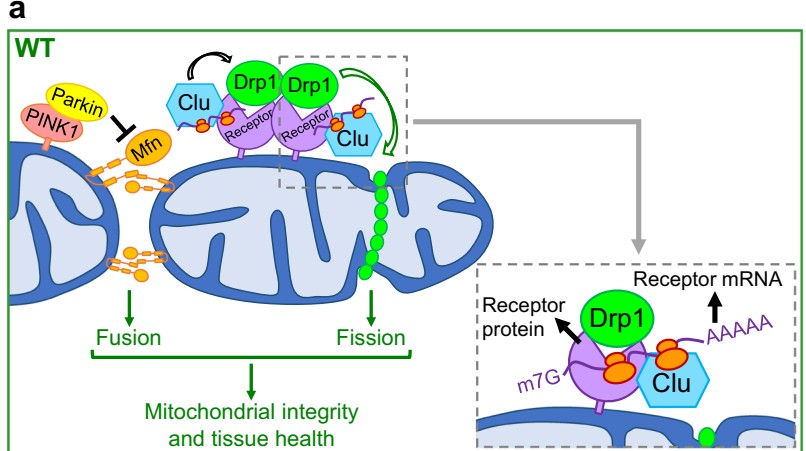
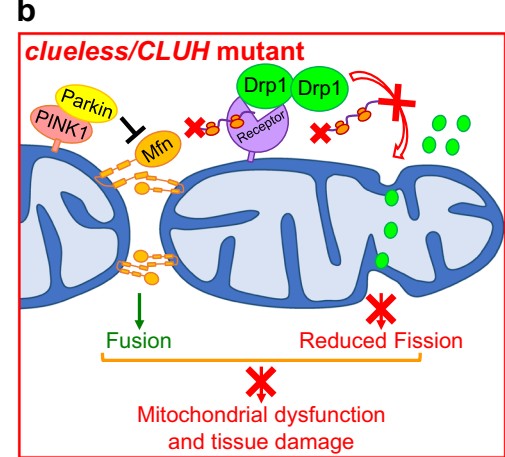

**Fig. 9 Models for how *clu/CLUH* promotes mitochondrial fission through regulating Drp1, and how functions of *clu-drp1* interact with those of *PINK1-parkin-mfn*. a** In wild-type cells, *Drosophila* Clueless and mammalian CLUH promote mitochondrial fission through regulating the recruitment of Drp1 from the cytosol to the OMM. In mammalian cells, this is accomplished by CLUH directly binding both mRNA and protein for Drp1 receptors Mff and MiD49. Moreover, CLUH regulates Mff and MiD49 protein levels through translational control. In *Drosophila* and mammalian cells, PINK1 and Parkin inhibit mitochondrial fusion by promoting degradation of Mfn proteins (*Drosophila* Marf, mammalian homologs Mfn1 and Mfn2). Together, the two pathways, *clu-drp1* and *PINK1-parkin-mfn*, balance the opposing actions of mitochondrial fission and fusion and maintain mitochondrial integrity and tissue health. **b** Loss of *clueless/CLUH* results in decreased levels of Drp1 receptor proteins due to dysregulated translation. This further leads to decreased recruitment of Drp1 onto the OMM, reduced mitochondrial fission, mitochondrial dysfunction, and tissue damage.

nuclear-encoded, mitochondria-destined proteins, and regulate translation and stability of these mRNAs[39,42,43,45]. CLUH can also form RNP particles with these bound mRNAs. These particles are thought to function as compartments within which CLUH regulates translation of bound mRNAs to modulate metabolic adaptation[45,46]. It will be interesting to further investigate how CLUH-bound *Mff* and *MiD49* mRNAs are distributed and organized in cells, and whether they enter and exit CLUH RNP particles under different physiological conditions.

Drp1 and Mff have also been shown to regulate peroxisome fission. However, we did not observe significant morphological changes of peroxisomes in *CLUH* KO cells as compared to control cells (Supplementary Fig. 8), despite decreased Mff levels and reduced recruitment of Drp1 to mitochondria in *CLUH* KO cells. This indicates that CLUH may only regulate mitochondria-localized Mff and recruitment of Drp1 to mitochondria, but not peroxisome-localized Mff or Drp1.

In *Drosophila*, *clu* null mutant phenotypes are similar to those of *PINK1* or *parkin* null mutants[37] (also this work). Previous studies have shown that *clu* OE suppresses phenotypes due to the lack of *PINK1* but not *parkin*[47,48]. These observations prompted a model that *clu* functions downstream of *PINK1* but upstream of *parkin* in the same genetic pathway[47,48]. In contrast, when we brought about an increase in *clu* expression levels using a stronger driver, Mef2-GAL4, *clu* OE suppresses phenotypes of *parkin* null mutants, in addition to those of *PINK1* null mutants in *Drosophila*. Therefore, our results suggest that *clu* functions either downstream of or in parallel to *PINK1-parkin*. To distinguish between these possibilities, we generated *clu PINK1* and *clu parkin* double mutants. While either *PINK1* or *parkin* single null mutants are viable, and *clu* null mutants can live up to 6 days, *PINK1 clu* and *parkin clu* double null mutants are lethal. Notably, *drp1* mutation shows similar patterns of synthetic lethal interactions with loss of *PINK1/parkin*. Flies bearing a *drp1* null allele (*drp1* null/+) in the *PINK* or *parkin* null mutant background are lethal, although *drp1* null/+ flies are viable[22,23]. Finally, *drp1* OE with Mef2-GAL4 also suppresses phenotypes due to the lack of either *PINK1* or *parkin*[22]. Together, these results allow us to propose an alternative model of how *clu* interacts with the

*PINK1-parkin* pathway, in which *clu-drp1* functions in parallel to *PINK1-parkin*.

The rescue of adult lethality of *clu* null mutants by *drp1* OE is significant, albeit partial. Partial rescue is commonly seen in in vivo genetic interactions, such as in our study of the *PINK1-parkin* pathway[20] and in previous studies of the *Numb*[66,67] and *Notch*[68,69] signaling pathway, and does not negate the strength of the genetic interactions in a pathway. In this *clu-drp1* interaction setting, this partial rescue could reflect the fact that *clu* has other downstream targets in addition to *drp1*, and/or it could be due to the tissue and temporal specificity of the in vivo *drp1* OE with Mef2-GAL4.

Previously, we and others found that Parkin negatively regulates levels of Mfn as an E3 ubiquitin ligase[22,24,70]. Overexpression of *mfn*, but not loss of *drp1*, phenocopies *PINK1* null and *parkin* null mutants in *Drosophila*[24], and *mfn* RNAi rescues either *PINK1* or *parkin* null mutant phenotypes[22,24]. A previous study suggested that *clu* promotes Mfn degradation in *Drosophila*[48]. We, however, did not detect changes of levels of Mfn proteins in *Drosophila* or mammalian cells, in response to either loss-of-function or OE of *clu/CLUH*. Furthermore, in contrast to *drp1* OE, *mfn* RNAi did not rescue the adult lethality or muscle degenerative phenotypes of *clu* null mutants in *Drosophila*. These results further strengthen our conclusion that *clu* functions to regulate *drp1* but not *mfn*. Taken together, our model posits that Clu/CLUH and PINK1-Parkin control the opposing actions of mitochondrial fission and fusion, by regulating Drp1 and Mfn, respectively. These two pathways, *clu-drp1* and *PINK1-parkin-mfn*, are crucial for maintaining mitochondrial quality control and tissue health (Fig. 9a, b).

PINK1 and Parkin promote mitophagy in mammalian cells. PINK1-mediated phosphorylation and Parkin-mediated ubiquitination of Mfn1 and Mfn2 promote the proteasomal degradation of Mfn1 and Mfn2, which prevents damaged mitochondria from fusing with healthy mitochondria[26,27]. Drp1 is recruited onto mitochondria during mitophagy to drive fission, resulting in the separation of one daughter mitochondrion that has depolarized membrane potential, from the other daughter mitochondrion that retains normal membrane potential[8]. Subsequently, the daughter

mitochondrion with depolarized membrane potential is eliminated through mitophagy. The function of Drp1 during mitophagy is independent of PINK1 and Parkin[13]. It will be interesting to further investigate whether CLUH plays a role in mitophagy, and if so, whether this function is related to Drp1.

Defects in mitochondrial fission have been shown to underlie the pathogenesis of multiple diseases. Mutations in the human drp1 (DNM1L) gene have been shown to cause neonatal lethality[71], childhood epileptic encephalopathy[72], refractory epilepsy[73–75], and dominant optic atrophy[76]. Mutations in Mff also cause childhood epileptic encephalopathy[77]. Recent studies identified two de novo missense mutations in the human CLUH gene, each altering an evolutionarily conserved amino acid in the CLUH protein[78,79]. These mutations cause congenital heart disease with neurodevelopmental disorders, including cognitive, motor, social, and language impairments. It will be interesting to determine whether defective mitochondrial fission, decreased mitochondrial recruitment of Drp1, disrupted interactions with Drp1 receptors and reduced Drp1 receptor levels underlie the pathogenesis of the CLUH-related diseases. Studies of Drp1 and CLUH could suggest novel therapeutic interventions for these diseases.

## Methods

**Molecular cloning for Drosophila experiments**. To generate UASt-clueless, EST clones RH51925 and GM10569 were obtained from the Drosophila Genome Research Center (DGRC). RH51925 contains a 3-nucleotide deletion (c.1483_1485delCTG) and several point mutations. GM10569 is a truncation without mutations, which only covers the C-terminal region of clueless. An N-terminal clueless fragment was generated by PCR of RH51925 (with the 3-nucleotide deletion). A C-terminal clueless fragment was generated by PCR of GM10569 so that all the point mutations in RH51925 were avoided. The full-length clueless coding sequence (CDS) was generated by bridging PCR, with the above N-terminal and C-terminal clueless DNA fragments as two template sequences. The full-length clueless CDS was then subcloned into the pENTR vector. The 3-nucleotide deletion was corrected using site-specific mutagenesis (Agilent Technologies, Santa Clara, CA, USA), which resulted in a pENTR-clueless construct with no mutations in the clueless CDS. This full-length clueless segment was subsequently cloned into the PTW or PTWF vector using the gateway cloning system (Invitrogen | Thermo Fisher Scientific, Carlsbad, CA, USA). The final constructs were sequenced to ensure the accuracy of the plasmids.

**Molecular cloning for mammalian cell experiments**. Full-length CLUH DNA was amplified from purchased EST clones and then subcloned into the vector pRK5-FLAG. Although no full-length CLUH EST was available, two overlapping EST sequences (Supplementary Table 3) were PCR amplified and ligated together via an internal BclI restriction site to obtain full-length CLUH DNA. Full-length MiD49 DNA was made through gene synthesis (GenScript Biotech, Piscataway, NJ, USA), and full-length MiD51 DNA was amplified from purchased EST clones (Supplementary Table 3). Both were then subcloned into the vector pRK5-Myc. Full-length human drp1 (DNM1L) DNA was subcloned into the PGW1 plasmid for mammalian expression of Drp1$^{WT}$. PGW1-Drp1$^{K38A}$ and PGW1-Drp1$^{S637A}$ were generated through site-directed mutagenesis[4,60]. All final constructs were sequenced to ensure the accuracy of the plasmids. GFP-Mff was obtained from Addgene (plasmid #49153).

**Molecular cloning for in vitro protein expression**. The pGEX6P1 (GE Healthcare) and pET28b (Novagen) vectors were generous gifts from Dr. Mark Arbing in the UCLA-DOE Institute. The modified pET21b+ vector contains a downstream PreScission Protease cleavage site, and the modified pET28a(+) vector contains a downstream PreScission Protease cleavage site as well as a GST tag[57]. Full-length human CLUH, drp1 isoform 1, drp1 isoform 1Δ514–639 (ΔInsertB), Mff isoform aΔ323–342, Mff isoform eΔ167–218 (ΔCCΔTM), Mff isoform eΔ199–218 (ΔTM), MiD49Δ1-51 and MiD49Δ1-124 were subcloned into the BamHI and EcoRI sites of the pGEX6P1 vector to generate N-terminal GST fusion proteins or subcloned into the NdeI and BamHI sites of the pET28b, modified pET21b+, or modified pET28a(+) vectors to generate His fusion proteins and C-terminal GST fusion proteins. All plasmids were verified by DNA sequencing.

**RNAi in mammalian cells**. Two stealth siRNAs targeted to CLUH were used to knock down CLUH (Supplementary Table 4) (Life Technologies, Carlsbad, CA, USA). Two Silencer siRNAs targeted to MiD49 (ID numbers: 129,182 and 37,021) and two Silencer siRNAs targeted to Mff (ID numbers: s533052 and s32461) were used to knock down MiD49 and Mff, respectively (Thermo Fisher Scientific). A

negative control siRNA (Silencer Negative Control No. 1 siRNA) was used to control for the effects of siRNA delivery (Thermo Fisher Scientific).

**Generation of CLUH KO HeLa cells using the CRISPR-Cas9 system**. Wild-type HeLa cells were a generous gift from Dr. Richard J. Youle. This HeLa cell line was originally acquired from the ATCC and authenticated by the Johns Hopkins GRCF Fragment Analysis Facility using STR profiling[80]. CLUH KO HeLa cells were generated through the CRISPR-Cas9 system[52,53], using this HeLa cell line. Four different spacers (Supplementary Table 5) targeting the second exon of CLUH were designed (http://crispr.mit.edu/) and cloned into the pX330-U6-Chimeric_BB-CBh-hSpCas9 vector (Addgene, Watertown, MA, USA), respectively. Each pX330-Spacer plasmid was individually transfected into HeLa cells using the X-tremeGENE™ nine DNA Transfection Reagent (Roche, Basel, Switzerland), following the manufacturer's instructions. Forty-eight hours after transfection, cells were trypsinized and re-plated into 48-well cell culture plates, with one single cell per well. These cells were cultured for 2 weeks until every single cell grew to a colony. Each single cell colony was then trypsinized and re-plated into 6-well cell culture plates to expand the cell population until it reached ~70% confluency. Each cell line was then screened for deletions in CLUH by genomic DNA extraction and PCR of a ~800 bp genomic region targeted by the spacers, followed by DNA sequencing (Supplementary Table 5). Two CLUH KO cell lines were isolated: CLUH-40 had a single nucleotide deletion and was resulted from pX330-Spacer-2 transfection; CLUH-43 had a 4-nucleotide deletion and was resulted from pX330-Spacer-3 transfection. The positions of the mutations in the CLUH gene are indicated in Fig. 5a. The absence of CLUH expression in both CLUH-40 and CLUH-43 HeLa cells was confirmed by western blotting and immunostaining using a rabbit anti-CLUH/eIF3X antibody (A301-765A, Bethyl Laboratories, Montgomery, TX, USA).

**RNA isolation, cDNA synthesis, and qPCR**. Total RNA was isolated from Drosophila thorax or mammalian cells using the Nucleospin RNA II kit (MACHEREY-NAGEL Inc.), followed by cDNA synthesis using the Clontech RNA to cDNA EcoDry Premix Kit (Takara Bio Inc.). Quantitative PCR was performed using the iTaq Fast Sybr Green enzyme mix (Bio-Rad Laboratories), 10 μL reactions in triplicate, on a LightCycler® 480 System (Roche) or a CFX Real-Time PCR System (Bio-Rad Laboratories). Data analysis was performed using the Bio-Rad CFX Maestro 1.1 software. Standard curves were generated for genes of interest and multiple housekeeping genes as controls, including Rpl32, eIF1α, alpha-Tubulin and GAPDH. Alternatively, semi-quantitative PCR was performed using a regular thermal cycler, and PCR products were visualized using agarose gels stained with DNA SafeStain (LAMDA Biotech). Fiji/ImageJ (National Institutes of Health, USA) was used to quantify the intensity of DNA bands on the gel images, with levels of genes of interest normalized to multiple control genes as indicated above.

**Drosophila strains and genetics**. The CaSpeR-FLAG-FlAsH-HA-drp1 strain was a generous gift from Dr. Hugo J. Bellen[49]. To generate UAS-clu and UAS-FLAG-clu transgenic flies, PTW/PTWF-clu constructs were injected into w[1118] flies (Rainbow Transgenic Flies, Inc, Camarillo, CA, USA). Multiple independent fly strains were collected and analyzed, and OE of clu was confirmed with RT-qPCR and western blotting. The clu[f04554] null mutant strain and the clu[d00713] hypomorph mutant strain were obtained from the Exelixis Collection at the Harvard Medical School. The UAS-clu RNAi strains were obtained from Vienna Drosophila Resource Center (VDRC). PINK1[5], parkin[25], dpk[21], UAS-drp1, UAS-mfn RNAi, IFM-GAL4, and Mef2-GAL4 flies were generated/used in our previous work[20,22,24,51,81]. Drosophila strains were maintained in a 25 °C humidified incubator. Transgenic flies were balanced against the same background: w[1118]; Bl/CyO; TM2/TM6B.

**Generation of mitotic clones using the FLP/FRT system in Drosophila**. yw, hsFLP70 (BDSC #6420) and yw; neoFRT42D (BDSC #5616) strains were obtained from the Bloomington Drosophila Stock Center (BDSC). The neoFRT42D allele was recombined with clu[f04554] and ubi-mRFP.nls, respectively. The recombined neoFRT42D clu[f04554] allele was further double-balanced with the CaSpeR-FLAG-FlAsH-HA-Drp1 allele. The recombined neoFRT42D ubi-mRFP.nls allele was further double-balanced with the yw, hsFLP70 allele. Female yw, hsFLP70/FM6; neoFRT42D ubi-mRFP.nls/CyO flies were crossed with male neoFRT42D clu[f04554]/CyO; CaSpeR-FLAG-FlAsH-HA-Drp1/TM6B flies to obtain hsFLP70/+; neoFRT42D clu[f04554]/neoFRT42D ubi-mRFP.nls; CaSpeR-FLAG-FlAsH-HA-Drp1/+ female progeny for measuring Drp1 levels in ovarian nurse cells. To induce mitotic recombination, the third-instar larvae were placed into a 37 °C water bath for heat-shock induction of FLP expression (5, 6, 7 days after setting up the cross, heat-shock for 2 h on each day). Virgin female flies were selected upon eclosion and fed with regular food plus extra yeast for 20 h to promote oogenesis. Ovaries were then dissected and fixed in 3.7% paraformaldehyde in phosphate-buffered saline (PBS) for 45 min and washed with PBS 3 times before proceeding to immunostaining.

**Drosophila longevity assay**. Newly eclosed flies were collected, with 10–15 flies in each vial. Females and males were placed in separate vials, with a total of >200 flies assayed for each genotype. Flies were maintained in a 25 °C humidified incubator,

scored for survival and death every day, and transferred to fresh food every 3 days[20].

**Drosophila ATP assay**. ATP levels were measured using lysates of five 2-day-old flies, with the ATP Bioluminescence Assay Kit HS II (Roche). For each genotype, ATP levels were normalized to total protein concentration, which indicates the total body mass of the flies. Total protein concentration was measured using the Pierce™ BCA Protein Assay Kit (Thermo Fisher Scientific, Waltham, MA, USA). Experiments were performed in triplicate and mean ± SEM was calculated.

**Drosophila TUNEL assay**. Terminal deoxynucleotidyl transferase dUTP nick end labeling (TUNEL) assays were carried out using adult male flies aged for 7 days at 25 °C. Thoraces of the flies were dissected and fixed in 4% paraformaldehyde in PBS. Muscle fibers were dissected and subsequently permeabilized and blocked in T-TBS-3% BSA [T-TBS: 0.1% Triton X-100, 50 mM Tris-Cl (pH 7.4), 188 mM NaCl][24]. After blocking, TUNEL staining was carried out using the In Situ Cell Death Detection Kit (Roche).

**Embedding, sections, Toluidine blue staining, and TEM**. Thoraces of 2-day-old flies were fixed in fixatives (1% paraformaldehyde, 1% glutaraldehyde, 0.1 M Phosphate Buffer), postfixed in 1% osmium tetroxide, dehydrated in gradient ethanol (50%, 70%, 100% sequentially), and embedded in Epon 812. After polymerization of Epon 812 at 65 °C overnight, blocks were cut to generate 1.5-μm thick sections using a glass knife, or 80-nm thin sections using a diamond knife (DiATOME, Hatfield, PA, USA) on a microtome (Leica, Wetzlar, Germany) (courtesy of Dr. Frank Laski at UCLA). The 1.5-μm thick sections were stained with Toluidine blue and examined using a regular light microscope (Zeiss, Oberkochen, Germany). The 80-nm thin sections were stained with uranyl acetate and lead citrate and examined using a JEOL 100 C transmission electron microscope (UCLA Brain Research Institute Electron Microscopy Facility)[24,51]. At least six samples from three different thoraces were examined for each genotype.

**Drosophila S2 cell culture and transfection**. S2 cells were cultured in Schneider's Medium with 1% penicillin/streptomycin and 10% fetal bovine serum (FBS) (Gibco). Cells were seeded 20 hours before transfection. Transfections were performed using the Effectene Transfection Kit (Qiagen, Valencia, CA, USA). After transfection, the cultured cells were incubated for 2–3 days before cell lysis and protein extraction.

**Mammalian cell culture and transfection**. HeLa cells (a generous gift from Dr. Richard Youle) were cultured and maintained with high glucose (4500 mg/L glucose) Dulbecco's Modified Eagle Medium (DMEM) supplemented with 1% penicillin/streptomycin and 10% FBS (Gibco). The cells were incubated at 37 °C and supplied with 5% CO$_2$ and passaged at ~70–80% confluency. To detach the cell monolayer, 2.5 mL of 0.25% trypsin with chelating agent EDTA (Gibco) was added for 5 min. The cells were then resuspended to 10 mL final volume, and a 1:20–1:5 dilution was made for maintenance. Plasmids were transfected into cells using the X-tremeGENE™ nine DNA Transfection Reagent (Roche). For RNAi experiments, siRNAs targeted to *CLUH*, *Mff*, *MiD49*, or the negative control siRNA were diluted with Opti-MEM (Gibco) and transfected into cells using Lipofectamine RNAiMAX (Thermo Fisher Scientific).

**Mammalian cell fractionation**. Totally, 5–10 × 10 cm dishes of HeLa cells were cultured, and transfections were carried out as indicated. When confluent, the media was aspirated and cells were washed twice with PBS. Using a cell scraper, the cells were detached in a small volume of PBS and pipetted into a microcentrifuge tube. The cells were then pelleted by spinning at 2800 g for 3 min in a benchtop centrifuge. The supernatant was discarded and the pellet re-suspended in 2.5–5 mL of cellular fractionation buffer, consisting of 20 mM HEPES (American Bioanalytical, Natick, MA, USA), 100 mM EGTA (Sigma-Aldrich), 75 mM sucrose (Sigma-Aldrich), and 225 mM mannitol (Sigma-Aldrich). Cells were incubated for 5 min and then homogenized by a motorized Potter-Elvehjem Tissue Homogenizer (Thomas Scientific, Swedesboro, NJ, USA) with 40–50 up and down strokes. Nuclei and unbroken cells were pelleted by two centrifugations for 5 min at 600 g, 4 °C. The resulting post-nuclear supernatant was centrifuged twice for 10 min at 7000 g, 4 °C, then once for 10 min at 10,000 g at 4 °C to obtain a mitochondria-rich pellet. The pellet was then lysed in 50-200 μL lysis buffer. To obtain the cytosolic fraction, the post-mitochondrial supernatant was transferred to an 11 × 34 mm polycarbonate tube (Beckman Coulter Life Sciences, Indianapolis, IN, USA), centrifuged at 21,000 g, 30 min, 4 °C in a Beckman Coulter Optima MAX ultracentrifuge (TLA-120.2 rotor). The resultant cytosolic supernatant was collected.

**Preparation of Drosophila lysates, protein extraction from cultured cells, and immunoprecipitation**. Thoraces from 6 to 10 adult flies were homogenized in RIPA buffer (0.5 M Tris-HCl, pH 7.4, 1.5 M NaCl, 2.5% deoxycholic acid, 10% NP-40, 10 mM EDTA) containing protease inhibitors (Roche). Total protein concentration was measured using the QuickStart Bradford assay kit (Bio-Rad Laboratories). Using a cell scraper, *Drosophila* S2 cells or mammalian cells were

detached in a small volume of PBS and pipetted into a microcentrifuge tube. The cells were then pelleted using a benchtop centrifuge. The supernatant was discarded and the pellet was lysed in RIPA buffer containing protease inhibitors (Roche). Cells were incubated on ice for 30 min and then centrifuged at 12,000 g at 4 °C for 10 min. The supernatant was transferred to a fresh microcentrifuge tube, and total protein concentration was measured using the QuickStart Bradford assay kit (Bio-Rad Laboratories). The samples were directly used for immunoprecipitation or western blotting or placed at −20 °C for short-term storage, or at −80 °C for long-term storage. Immunoprecipitation was performed with the lysates from *Drosophila* S2 cells or mammalian cells, using Dynabeads Protein G (Thermo Fisher Scientific). Proteins bound to beads were eluted in SDS sample buffer (Bio-Rad Laboratories) containing 2-Mercaptoethanol. Cell lysates and immunoprecipitates were analyzed by western blotting, using the same amount of total protein from flies or cells of different genotypes.

**Antibodies used for immunoprecipitation (IF), western blotting (WB), immunoprecipitation (IP), and proximity ligation assay (PLA)**. The following primary antibodies were used: mouse anti-HA (Millipore, Burlington, MA, USA) (IF 1:200, WB 1:1000, IP 1:300), rabbit anti-HA (Millipore) (IF 1:200, WB 1:1000), mouse anti-Myc (Millipore) (IF 1:200, WB 1:1000, IP 1:300), rabbit anti-Myc (Cell Signaling Technology, Danvers, MA, USA) (IF 1:200, WB 1:1000), mouse anti-FLAG (GenScript Biotech, Piscataway, NJ, USA) (IF 1:200, WB 1:1000, IP 1:300), mouse anti-FLAG (clone M2, Sigma-Aldrich, St. Louis, MO, USA; GenScript Biotech, clone 5A8E5) (IF 1:200, WB 1:1000, IP 1:300), rabbit anti-FLAG (GenScript Biotech) (IF 1:200, WB 1:1000), mouse anti-GFP (clone GFP-20, ascites fluid, Sigma-Aldrich) (IP 1:300), rabbit anti-GFP (Thermo Fisher Scientific) (IF 1:200, WB 1:1000), rabbit anti-CLUH/eIF3X (A301-765A or A1259-1309A, Bethyl laboratories) (IF 1:1000, WB 1:1000, PLA 1:100), rabbit anti-CLUH (Aviva Systems Biology, ARP70642_P050) (PLA 1:100), mouse anti-Drp1 (Abcam) (IF 1:200, WB 1:1000, PLA 1:100), rabbit anti-*Drosophila* Marf (a generous gift from Dr. Alexander J. Whitworth) (WB 1:1000), mouse anti-Mfn1 (Abcam) (WB 1:1000), rabbit anti-Mfn2 (Proteintech Group, Inc, Rosemont, IL, USA) (WB 1:1000), mouse anti-ATP Synthase (MitoSciences, Eugene, OR, USA) (IF 1:200), mouse anti-TOM20 (BD Transduction Laboratories, San Jose, CA, USA) (IF 1:300), goat anti-Hsp60 (Santa Cruz Biotechnology, Dallas, TX, USA) (IF 1:200), mouse anti-Porin (MitoSciences Inc., Eugene, OR, USA) (WB 1:1000), rabbit anti-VDAC1/Porin (Abcam) (WB 1:1000), rabbit anti-Actin (Sigma-Aldrich) (WB 1:1000), mouse anti-Tubulin (Sigma-Aldrich) (WB 1:1000), mouse anti-Puromycin (Kerafast, 3RH11) (IF 1:1000, PLA 1:500), rabbit anti-Mff antibody (Proteintech, 17090-1-AP) (IF 1:200, PLA 1:100), rabbit anti-MiD49/SMCR7 (Thermo Fisher Scientific, PA5-99984) (IF 1:50, PLA 1:50–1:200). The following secondary antibodies were used for WB: ECL anti-rabbit IgG HRP-linked whole antibody from the donkey, ECL anti-mouse IgG HRP-linked whole antibody from sheep (GE Healthcare) (1:10,000). The following secondary antibodies were used for IF: Alexa Fluor 488 donkey anti-mouse IgG, Alexa Fluor 488 goat anti-mouse IgG, Alexa Fluor 546 donkey anti-mouse IgG, Alexa Fluor 488 donkey anti-rabbit IgG, Alexa Fluor 488 goat anti-rabbit IgG, Alexa Fluor 546 goat anti-rabbit IgG, Alexa Fluor 488 donkey anti-goat IgG, Alexa Fluor 594 donkey anti-goat IgG (Thermo Fisher Scientific) (1:500).

**Immunohistochemistry and immunocytochemistry**. For analysis of *Drosophila* muscle mitochondrial morphology, thoraces were fixed in 4% paraformaldehyde in 1× PBS. After thoraces were washed in 1× PBS twice, indirect flight muscle fibers were individually dissected and isolated. For analysis of ovarian nurse cells, freshly eclosed female flies were maintained on wet yeast paste for 24 hours prior to ovary dissection, and individual stage 10 egg chambers were dissected following fixation. Muscle fibers or egg chambers were then permeabilized and blocked in PBS + 0.1% Triton X-100 with 5% FBS (Gibco). Mammalian cells were fixed in 10% formalin for 10 min at 37 °C, permeabilized with PBS + 0.1% Triton X-100 for 15 min at room temperature, and then blocked in PBS + 0.1% Triton X-100 with 5% FBS for 1 hr at room temperature. For immunofluorescence of mitochondrially bound Drp1, cells were permeabilized in a low-concentration digitonin buffer [0.001% digitonin, 20 mM 4-(2-hydroxyethyl)-1-piperazineethanesulfonic acid (HEPES), 150 mM NaCl, 2 mM MgCl$_2$, 2 mM EDTA, 320 Mm sucrose, pH 7.4] for 90 sec at 37 °C and then immediately fixed[31]. Cells were then processed as described above. After tissue or cells were blocked with 5% FBS, immunostaining was carried out with primary antibodies in PBS + 0.1% Tween20 + 5% FBS. After tissue or cells were washed 3 times with PBS + 0.1% Tween20, secondary antibodies were used in PBS + 0.1% Tween20 + 5% FBS. Cells were washed 3 times with PBS + 0.1% Tween20 before mounting. Nuclear staining was performed with 1× Hoechst (Thermo Fisher Scientific) for 10 min at room temperature after secondary antibody staining or with DAPI in the mounting media (SouthernBiotech™ Dapi-Fluoromount-G™ Clear Mounting Media, Fisher Scientific).

**RNA immunoprecipitation (RIP)**. RIP was performed using the Magna RIP™ RNA-Binding Protein Immunoprecipitation Kit (Millipore Sigma). CLUH was immunoprecipitated using 5 μg of a rabbit anti-CLUH antibody (Aviva Systems Biology, ARP70642_P050). The control IgG was provided by the manufacturer and 5 μg was used for the control RIP. Immunoprecipitated RNAs were converted to

cDNAs through reverse transcription using the RNA to cDNA EcoDry Premix (TaKaRa). cDNAs were used as templates for PCR (Quick-Load Taq 2X Master Mix, New England Biolabs) to detect target genes. Results were analyzed using agarose gel electrophoresis, and gel images were taken using the KODAK Gel Logic 112 imaging system with the KODAK Molecular Imaging software.

**RNAscope.** Single-molecule RNA FISH (RNAscope) was carried out using the RNAscope Multiplex Fluorescent v2 kit (Advanced Cell Diagnostics). Target probes to detect human *MiD49* and *Mff* were designed and synthesized by the manufacturer. Cells were grown as indicated, harvested at 75% confluency, fixed in 10% formalin, and pretreated according to the manufacturer's instructions. The Signal was amplified according to the manufacturer's instructions and with the Opal 570 dye (Akoya Biosciences). Mitochondria were stained using a goat anti-Hsp60 antibody (Santa Cruz Biotechnology) combined with Alexa Fluor 488 donkey anti-goat secondary antibody (Thermo Fisher Scientific).

**Puromycylation-PLA (Puro-PLA).** HeLa cells were incubated with 3 μM puromycin for 10 min in the full medium at 37 °C. Incubation was stopped by two fast washes in prewarmed PBS-MC (1× PBS, pH 7.4, 1 mM MgCl$_2$, 0.1 mM CaCl$_2$) and cells were fixed for 20 minutes in 4% PFA-sucrose (4% paraformaldehyde diluted from 20% paraformaldehyde, Electron Microscopy Sciences; 4% sucrose in PBS-MC) at room temperature. After fixation, cells were washed, permeabilized with 0.5 % Triton X-100 in 1× PBS (pH 7.4) for 15 min, and blocked with blocking buffer (5% FBS in 1× PBS) for 1 h. Newly synthesized proteins were detected using a mouse anti-puromycin antibody in combination with a rabbit anti-Mff or anti-MiD49 antibody as primary antibody pair (details and dilutions described above in the "Antibodies used for IF, WB, IP and PLA" paragraph). Experiments were performed using Duolink™ In Situ Red Starter Kit Mouse/Rabbit (Sigma) according to the manufacturer's protocols. Mitochondria were stained using a goat anti-Hsp60 antibody (Santa Cruz Biotechnology).

**Confocal imaging and image analysis.** Images were acquired using a Zeiss LSM 5 confocal microscope (UCLA Brain Research Institute), a Zeiss LSM 700 confocal microscope, or a Zeiss LSM 880 with Airyscan (the Broad Stem Cell Research Center at UCLA), with the Zeiss ZEN software (black edition). All image analysis was performed using the Fiji/ImageJ software (National Institutes of Health, USA). For mitochondrial morphology quantification in HeLa cells, analysis was limited to regions of interest of cells, where individual mitochondria could be visualized with high resolution. For measurement of mitochondrially localized Drp1 fluorescence intensity, HeLa cells were briefly treated with 0.001% digitonin before fixation as described above in the "Immunohistochemistry and immuno-cytochemistry" paragraph, in order to reduce the level of cytosolic Drp1 and to improve visualization of mitochondrial Drp1. The fluorescence intensity of mito-chondrially bound Drp1 was analyzed by first creating a binary mask of the mitochondrial channel (using the anti-Hsp60 immunofluorescence signals). This was then used to subtract all extramitochondrial Drp1 fluorescence, and the remaining fluorescence intensities of Drp1 were measured[31,57].

**Statistics and reproducibility.** The overall study design was a series of controlled laboratory experiments using *Drosophila* and cultured mammalian cells, as described in detail in the Figure legends and Methods. All experiments were repeated at least three individual times with similar trends. For in vivo experiments, the number of flies used for each control or experimental group was indicated in the Figure legends as well as in the above Method sections. For experiments using cultured cells, the total number of cells counted for each control or experimental group was indicated in the Figure legends. For images analysis, regions of interest for each control or experimental group were randomly assigned for image acquisition. All quantitative experiments were evaluated for statistical significance using one-way ANOVA with post hoc Tukey's HSD test, or two-sided Student's *t*-tests. Means ± SEM, the corresponding data points (as dot plots), and the exact *P*-values are displayed in the Figures. $P < 0.05$: significantly different from the control group; $P > 0.05$: not significant. Data were plotted using the GraphPad Prism v.9.2.0 software.

**Reporting summary.** Further information on research design is available in the Nature Research Reporting Summary linked to this article.

## Data availability

All data are available within the Article, Supplementary Information, or Source Data file. Source data for Fig. 3a, b, and Supplementary Table 1 have been provided as Supplementary Table 2. All other source data are provided in the Source Data file with this paper. The knockout cell lines and transgenic flies generated in this work are available from the corresponding author upon request. Source data are provided with this paper.

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

## Acknowledgements

We thank H.J. Bellen, A. Whitworth, and R.J. Youle for reagents, A.M. van der Bliek for comments on the manuscript, Y. Sun and P. Zheng for technical advice, and F.A. Laski, L. Dreier, E. Videlock, the BRI/UCLA EM Core Facility and the BSCRC/MCDB Microscopy Core Facility at UCLA for equipment. This work was supported by the China Scholarship Council Fellowship and UCLA Dissertation Year Fellowship to H.Y., the Wellcome Trust-NIH Ph.D. Studentship (WT088328AIA) to C.S., C.B., and R.J.H., the Intramural Research Program of the NINDS to C.B., and the R01 from the National Institute on Aging, Glenn Foundation for Medical Research, Kenneth Glenn Family Foundation, the Louis B. Mayer Foundation, the B. Freeman and R. Spogoli Fund for Aging and Neurodegeneration, the UCLA Laurie and Steven Gordon Commitment to Cure Parkinson's Disease, the Renee and Meyer Luskin Family Fund to M.G.

## Author contributions

H.Y. designed, performed, and analyzed experiments and wrote the paper. C.S., R.L., J.Y., C.B., and R.J.H. designed, performed, and analyzed experiments. B.A.H. designed and analyzed experiments and wrote the paper. D.C.C. designed and analyzed experiments. M.G. designed and supervised the experiments, analyzed the results, provided the funding, and wrote the paper.

## Competing interests

The authors declare no competing interests.
