## [Peer Review File · Nature Communications]

Clueless/CLUH regulates mitochondrial fission by promoting recruitment of Drp1 to mitochondriaREVIEWER COMMENTS

Reviewer #1 (Remarks to the Author):

In this manuscript, Yang et al describe the involvement of CLUH in mitochondrial fission. Through work in drosophila, they demonstrate that altered expression of clu affected mitochondria morphology in a manner that resembles fission/fusion factors. I.e., overexpression leads to mitochondria fragmentation and RNAi leads to elongated mitochondria. Interestingly, overexpression of the fission factor Drp1 leads to partial rescue of the mitochondrial phenotypes in Drosophila muscle. Complementary, overexpression of clu enhances the recruitment of Drp1 to mitochondria, and enhances fission. The authors then tested these roles in mammalian cell line (HeLa), and also in this system, CLUH enhances Drp1 recruitment to mitochondria. Finally, the recruitment in HeLa was found to involve the Drp1 receptors MiD49 and Mff, by improving their binding to Drp1 (as demonstrate in vitro) and/or by increasing their translation (as evident from western and RT-PCR analyses). Overall, this is a very significant work, which provide new insights to the function of CLUH and to mechanisms of mitochondrial fission. The parallel work in fly and human cell line further substantiates the relevance of the findings.

Comments:

1. Some of the experiments are missing quantifications and repetitions. In particular the IP experiments (Fig 4G, H (it is actually indicated that these were done in triplicate, yet no quantification is presented), Fig 6 A-C). Also, the IP panels are usually missing a control of an irrelevant protein (e.g. Actin) to confirm that binding is specific to the protein of interest. Also some of the imaging analyses are missing quantification (Fig 6G,H, 7F-H)
2. Some of the results are shown only in the HeLa system, while others only in the fly. It will be much more compelling to show main findings in both systems. Specifically, the role of Drp1 receptors was not examined in fly:
 - a. In Fig. 3, it appears that Drp1 overexpression is leading to only a partial rescue. Could overexpression of the receptors improve that?
 - b. In the IP experiments in Fig 4G,H, can the authors test if the receptors are also present by using the relevant antibodies?
 - c. Complementary, overexpression of Drg1 in CLUH KO HeLa cells should be tested, to show rescue of the clustering phenotype.
3. The interaction between CLUH and the receptor is presented primarily in vitro. I think that this should be substantiated by some in vivo analysis, for example by knock down of receptors followed by IP co-IP of CLUH and Drg1 (supposedly should be diminished.)
4. The possible impact on localized translation of Mff and MiD49 should be substantiated, possibly by assays like PLA or TRICK. Also, ribosome-proximity is an option – this was done recently by Vardi-Oknin

and Arava. Since they showed that CLUH has a role in localized translation in mammalian cell line, this work is relevant also to the presented model.

5. The physiological significance of the proposed interaction is not clear (at least for the HeLa cells). One possibility is to look at stress conditions in which CLUH levels increase, and check the impact on mitochondria fission (via Mid49 of Mff, through their knock down). Since CLUH was shown to affect mRNAs encoding OXPHOS components, maybe conditions that are related to this will be relevant.

6. For the experiment with Mff and MiD sequestration to the lysosome, it will be informative to present the impact of sequestration on mitochondria morphology. Presumably, mitochondria fission should be affected.

Reviewer #2 (Remarks to the Author):

Clueless (Clu/CLUH) is an RNA-binding protein with important functions in regulating mitochondria morphology and distribution. Drp1 is a key regulator of mitochondrial fission. In this work, the authors interrogate the relationship between Clu and Drp1 using *Drosophila* and cultured cell models. Clu loss-of-function and over-expression phenotypes are described in the *Drosophila* muscle and additionally in the context of mitochondrial quality-control mutants (i.e., pink and parkin mutants). Gross morphological changes in mitochondria shape and distribution following loss or overexpression of mammalian CLUH are similarly presented. Because overexpression of drp1 partially rescues the clu null phenotype, Clu is suggested to function upstream of Drp1 activity. Consistent with this idea, drp1 over-expression seems to increase Drp1 localization to mitochondria. Co-IP data indicate Clu and Drp1 associate biochemically in *Drosophila* extracts. Further analysis suggests the interaction between CLUH and Drp1 is indirect and may be mediated by OMM receptors, Mff and MiD49. Intriguingly, manipulation of CLUH dosage appears to alter steady state levels of Mff and MiD49 protein, although mechanisms underlying this response remain unclear.

Overall, the authors present a body of data nicely implicating a genetic and biochemical relationship between Drp1 and Clu. The work is of significance given implications to mitochondrial integrity and quality, so essential to health and development. Nevertheless, failure to display quantification of key findings makes it difficult to interpret the work or determine veracity of the stated conclusions. Finally, the relationship between Clu and Mff or MiD49 protein levels remains murky, so the model presented in Figure 8 seems highly speculative. No evidence indicating translational control is presented or discussed beyond changes in protein levels via western blot. I recommend revision prior to publication.

Major Points

1. The authors nicely present quantification of some data (e.g. Supp. Fig 1F and Fig 2G present quantification of mitochondrial size), however it is imperative that the same rigor of quantification is presented throughout the manuscript. I also discourage the use of bar charts for most of these measurements; showing individual data points through scatter plots is recommended as it helps the reader see if the data follow a given trend, if there are significant outliers, and also how many samples were quantified. Quantification is required for the following:

Fig 1 A-L

Fig 3C-E''

Fig 4 D-F''

Fig 5 C-I'

Fig 6 G-h''

Fig 7 F-H'' and Fig 7 K (Show RNA levels quantified across replicates).

Additionally, please indicate N number of cells examined per experimental replicate for all panels and how many cells displayed a given phenotype.

2. How Clu/CLUH regulates Mff or MiD49 protein levels remains unclear. Further unclear is whether Mff and MiD49 are subject to the same regulatory process, as over-expression of CLUH results in an increase of MiD49 protein levels, while loss of CLUH leads to a corresponding loss of MiD49. The results for Mff are far more subtle, although not discussed by the authors. Statistically, over-expression of CLUH does not alter Mff, while loss of CLUH leads to a decrease (but not loss) of Mff. These distinctions between Mff and MiD49 responses are likely biologically relevant and should be noted.

3. What contributes to the decrease of Mff or MiD49 proteins upon loss of CLUH is not addressed. While the authors present in Figure 7K a semi-quantitative RT-PCR gel and 7L shows a protein:RNA ratio, it is necessary to show replicated qPCR data of normalized Mff and MiD49 levels in WT vs CLUH KO vs CLUH OE samples. A protein:RNA ratio is not particularly helpful, as protein production does not necessarily scale linearly with RNA amount. The semi-quantitative gel does not seem to indicate loss or destabilization of Mff or MiD49 mRNAs in response to loss of CLUH; confoundingly, these mRNA levels actually seem higher, but a qPCR would show this more clearly. The authors should then provide interpretation of the RNA vs protein level data and discuss models.

4. The model presented in Figure 8 is speculative. There are various ways one can test translational status, such as by visualizing nascent translation in vivo (e.g. Puro-PLA or Sun-TAGs) or by examining ribosomal occupancy or distributions of proteins in polysome gradients, but these data are not presented here. While translational control is one model to account for reduced steady-state Mff or MiD49 proteins in the absence of CLUH, other models may be considered, such as protein destabilization/degradation.

Minor Points

1. Statistical analyses are presented throughout, although the authors should reconsider the use of the Student's t-test when making multiple comparisons in Figs 1, 3B, and 7. Likewise for Supp Fig 1B, 2A, 3K.

2. Rescue of *clu* null phenotypes by OE of *drp1* is convincing, albeit partial. The authors should note the rescue is partial or incomplete and acknowledge the idea that other factors also function downstream of Clu important for viability.

3. Does manipulation of Clu dosage in *Drosophila* (RNAi, mutant, or OE) alter Mff protein levels?

4. The authors use two independent techniques to show that loss of *clu* does not alter Drp1 protein levels (Fig 4A and D). It was surprising to note in Fig 4D, Drp1 signals seem relatively unaffected regardless of Clu dosage, given, in cultured cell models, the authors suggest that CLUH depletion may lead to a decrease in Drp1 recruitment to mitochondria (Fig 6E). Could the authors discuss this discrepancy?

In both *Drosophila* and cell lines, CLUH OE seems to increase overlap of mitochondria and Drp1 signals (Fig4F, Fig 6H – although both need quantification, perhaps through a Pearson's coefficient). Is one system more sensitive to Clu/CLUH loss, while both respond comparably to Clu/CLUH OE?

5. Through cell fractionation, the authors show CLUH OE leads to an increase in Drp1 levels in the mito fraction and a corresponding loss in the cytoplasm. These are nice data, but can the authors discuss why the recruitment of CLUH itself to the mito fraction seems relatively modest? Could this support the

hypothesis where CLUH itself is not directly responsible to recruit or drag Drp1 to mitochondria? (This would be consistent with data shown in Fig 7F–H”)

6. For cell culture work, CLUH CRISPR KO lines are compared to “WT” cells. Please define these controls. Are they non-edited CRISPR lines or other? One must consider potential for off-target effects.

7. In the absence of quantification, statements on lines 227 (“significant reduction”) and 250 (“significantly increased colocalization”) are misleading, as the data are not shown. These metrics need to be quantified before such qualitative statements can be made.

8. The data shown in Fig 6 D-F are not particularly compelling. Specifically, how the Drp1 fluorescence intensity at mitochondria was calculated is not clear. The authors may consider a more standard colocalization analysis, such as a Pearson’s or Manders correlation coefficient.

9. In the Methods (lines 379-90), please add dilutions of antibodies used for IF vs western blotting. These details are key to ensure reproducibility.

10. The TPR domain of Clu is important for RNA-binding:
<https://www.ncbi.nlm.nih.gov/pmc/articles/PMC4823986/>

Have the authors examined whether delta TPR mutants recapitulate mutant phenotypes? This line of experimentation would get to the question of mechanism, as related to possible contributions regulating Mff and MiD49.

11. Is there evidence for a role for Clu in RNA localization? Are Mff or MiD49 mRNAs localized appropriately in CLUH KO or OE sample?

Reviewer #3 (Remarks to the Author):

The manuscript by Yang H et al. entitled "Clueless/CLUH regulates mitochondrial fission by promoting recruitment of Drp1 to mitochondria" describes the role of the RNA-binding protein CLUH in Drp1-dependent mitochondrial fission using drosophila and mammalian cell culture as two model systems.

Strengths:

- The authors convincingly demonstrate the pivotal role of CLUH for mitochondrial morphology and mitochondrial function with implications for fitness, in their two model systems
- The authors provide a number of convincing pieces of evidence that CLUH functions on mitochondrion-localized Drp1, which enables mitochondrial fission

Limitations:

- Although the authors provide data suggesting that CLUH is involved in recruiting Drp1 to mitochondria, these data are comparatively weak, and the proposed mechanisms behind requires experimental validation (see below)
- The authors ignore the potential role of peroxisomes, which share parts of the fission machinery with mitochondria

Major points:

1.) The CLUH-dependent recruitment of Drp1 to mitochondria needs further experimental validation or the interpretation of existing data must be handled with more caution. The authors should perform at least two of the suggestions (or equivalent approaches) to substantiate their model (preferentially as described in d):

a.) Fig. 6D-F: The effects on mitochondrial Drp1 localization for CLUH ko are very moderate, I recommend cell fractionation experiments as shown in Fig. 6I with CLUH ko cells.

b.) Fig. 6G-I: The effects on mitochondrial Drp1 localization for CLUH OE appear substantial, but then fluorescence microscopy experiments as in Fig. 6D-F should result in similar effects. I recommend performing these experiments.

c.) Fig. 6: I recommend expressing CLUH in CLUH ko cells using a controllable promoter. Drp1-recruitment which depends on the expression level of CLUH would be more convincing. Alternatively, time-dependent recruitment of Drp1 upon CLUH expression using live-cell microscopy could be used.

d.) Fig. 7KL: The authors show that the steady state levels of the mRNA encoding the Drp1 receptors MiD48 and Mff are decreased in the absence of CLUH. However, it remains unclear whether CLUH binds and stabilizes these mRNAs in this context, and most importantly, it remains unclear whether this binding occurs at the OMM. Any experimental validation of the latter would help a lot to support the model provided by the authors.

2.) The model combines interpretations of data from flies and cell culture, and it is not clear which part of the model is validated by fly data only, by fly and cell culture data, and by cell culture data only. This should be indicated in the figure and the figure legend.

3.) The authors need to discuss the potential role of peroxisomes. Could CLUH promote Drp1-dependent fission of peroxisomes as well? The reviewer does not request experimental validation but peroxisomes should be at least mentioned in the discussion

Minor points:

1.) Fig. 5: Quantifications for changes in mitochondrial morphologies are recommended

Point-by-point response to the reviewers' comments

We appreciate the reviewers' suggestions on our manuscript revision. We were encouraged by the comments by Reviewer #1: *"this is a very significant work, which provide new insights to the function of CLUH and to mechanisms of mitochondrial fission"* and *"the parallel work in fly and human cell line further substantiates the relevance of the findings"*. We were also pleased that Reviewer #2 highlighted that: *"the authors present a body of data nicely implicating a genetic and biochemical relationship between Drp1 and Clu"* and *"The work is of significance given implications to mitochondrial integrity and quality, so essential to health and development"*. Lastly, we appreciated Reviewer #3 for his/her comments that: *"The authors convincingly demonstrate the pivotal role of CLUH for mitochondrial morphology and mitochondrial function with implications for fitness, in their two model systems"* and *"The authors provide a number of convincing pieces of evidence"*.

The reviewers raised several key concerns, which we will address point-by-point below. In particular, we performed several additional experiments to address the mechanism by which CLUH regulates Drp1 receptors MiD49 and Mff. We are delighted to report that we have demonstrated that CLUH regulates MiD49 and Mff levels through binding to their cognate mRNAs and promoting their translation. We are pleased that the reviewers made this suggestion, as these new findings significantly strengthen our paper. We have also rewritten a portion of the manuscript with all the modifications color tracked, and included the revised manuscript in the submission.

Reviewer #1 (Remarks to the Author):

In this manuscript, Yang et al describe the involvement of CLUH in mitochondrial fission. Through work in drosophila, they demonstrate that altered expression of clu affected mitochondria morphology in a manner that resembles fission/fusion factors. I.e., overexpression leads to mitochondria fragmentation and RNAi leads to elongated mitochondria. Interestingly, overexpression of the fission factor Drp1 leads to partial rescue of the mitochondrial phenotypes in Drosophila muscle. Complementary, overexpression of clu enhances the recruitment of Drp1 to mitochondria, and enhances fission. The authors then tested these roles in mammalian cell line (HeLa), and also in this system, CLUH enhances Drp1 recruitment to mitochondria. Finally, the recruitment in HeLa was found to involve the Drp1 receptors MiD49 and Mff, by improving their binding to Drp1 (as demonstrate in vitro) and/or by increasing their translation (as evident from western and RT-PCR analyses). Overall, this is a very significant work, which provide new insights to the function of CLUH and to mechanisms of mitochondrial fission. The parallel work in fly and human cell line further substantiates the relevance of the findings.

We thank Reviewer #1 for these very positive comments.

Comments:

1. Some of the experiments are missing quantifications and repetitions. In particular the IP experiments (Fig 4G, H (it is actually indicated that these were done in triplicate, yet no quantification is presented), Fig 6 A-C). Also, the IP panels are usually missing a control of an irrelevant protein (e.g. Actin) to confirm that binding is specific to the protein of interest. Also some of the imaging analyses are missing quantification (Fig 6G,H, 7F-H)

We thank Reviewer #1 for these thoughtful comments. We have quantified the IP experiments, and added the quantifications to the respective figures (*Drosophila*: Fig 4k, m; HeLa cells: Fig 6d and Fig 7b, d).

We are also in complete agreement with Reviewer #1 that IP panels *"need a control of an irrelevant protein to confirm that binding is specific to the protein of interest"*. In *Drosophila* S2 cells, we

showed that Clu does not bind VCP (Supp Fig 2b) and found that Clu does not bind Parkin (data not shown), confirming that Clu-Drp1 and Clu-PINK1 protein interactions (Fig 4) are specific. In each IP experiment, we used mCherry fused to the tag used as a negative control, to show that binding is specific rather than due to the tags used. In HeLa cells, we showed that CLUH does not bind Mfn2 (Supp Fig 4c) or MiD51 (Supp Fig 5a), which confirms that CLUH-Drp1, CLUH-MiD49, and CLUH-Mff protein interactions are specific.

In addition, we appreciate the suggestion regarding image quantification. We have added quantifications of co-localization using Pearson's coefficient to Fig 6i, o and Fig 7n.

2. Some of the results are shown only in the HeLa system, while others only in the fly. It will be much more compelling to show main findings in both systems. Specifically, the role of Drp1 receptors was not examined in fly:

a. In Fig. 3, it appears that Drp1 overexpression is leading to only a partial rescue. Could overexpression of the receptors improve that?

We thank Reviewer #1 for this comment. We completely agree that the rescue is partial. Meanwhile, we would like to note that most genetic interactions in *Drosophila* show partial rescue, even for genes in confirmed key pathways. For example, we showed that overexpression of *parkin* rescues the male sterility (62% rescue) and mitochondrial morphology defects of *PINK1* mutant flies (50% of the mitochondria with abnormal morphology were rescued) (Clark *et al.*, *Nature* 2006). Other examples include earlier studies on the *Notch* signaling pathway, where researchers showed that *Drosophila frizzled (fz)* and *dishevelled (dsh)* mutant eye phenotypes were partially rescued by overexpressing downstream components (Strutt, Weber and Mlodzik, *Nature* 1997; Cooper and Bray, *Nature* 1999). These partial rescues can reflect the fact that the pathways being studied have multiple downstream branches, or can be due to the tissue and temporal specificity of the controlled gene overexpression *in vivo*. However, partial rescue does not negate the strength of the pathway; instead, it indicates fairly robust genetic interaction.

We have also added to the text (Discussion, Lines 449-454) “The rescue of adult lethality of *clu* null mutants by *drp1* overexpression is significant, albeit partial. Partial rescue is commonly seen in *in vivo* genetic interactions, such as in our study of the *PINK1-parkin* pathway [20] and in previous studies of the *Numb* [65, 66] and *Notch* signaling pathway [67, 68], and does not negate the strength of the genetic interactions in a pathway. In this *clu-drp1* interaction setting, this partial rescue could reflect the fact that *clu* has other downstream targets in addition to *drp1*, and/or it could be due to the tissue and temporal specificity of the *in vivo drp1* overexpression with Mef2-GAL4.”

b. In the IP experiments in Fig 4G, H, can the authors test if the receptors are also present by using the relevant antibodies?

We would definitely like to examine whether Clu binds Drp1 receptors in *Drosophila*. Flies have the *Mff* gene, but not *MiD49* or *MiD51*. However, antibodies for *Drosophila* Mff are not available in the field, thus preventing us from performing the experiments.

To clarify this issue, we have modified the following sentences in the text (Lines 283-285): “*Drosophila* has Mff [28, 57], but not MiD49 or MiD51. The role of *Drosophila* Mff in recruiting Drp1 still remains unclear [28]. Therefore, we focused our studies of interactions between CLUH and Drp1 receptors using HeLa cells.”

c. Complementary, overexpression of Drp1 in CLUH KO HeLa cells should be tested, to show rescue of the clustering phenotype.

We appreciate Reviewer #1 for this great idea and have conducted this experiment. The results were as expected: overexpression of Drp1 indeed rescues the mitochondrial clustering phenotype in CLUH KO HeLa cells. We added this new data to Fig 5l-p.

3. The interaction between CLUH and the receptor is presented primarily in vitro. I think that this should be substantiated by some in vivo analysis, for example by knock down of receptors followed by IP co-IP of CLUH and Drp1 (supposedly should be diminished.)

Reviewer #1 asked us to substantiate our *in vitro* protein interaction results using *in vivo* analysis. We addressed this point accordingly and indeed, we demonstrated CLUH-Drp1 protein interactions via Proximity Ligation Assays (PLA) in HeLa cells. Furthermore, we found that knock down of *MiD49* results in decreased CLUH-Drp1 PLA signals, indicating that Drp1 receptors are required for the protein interactions between CLUH and Drp1 (see Fig 7h-j).

4. The possible impact on localized translation of Mff and MiD49 should be substantiated, possibly by assays like PLA or TRICK. Also, ribosome-proximity is an option – this was done recently by Vardi-Oknin and Arava. Since they showed that CLUH has a role in localized translation in mammalian cell line, this work is relevant also to the presented model.

This is an excellent suggestion. We addressed this point via the Puro-PLA assay, and found significantly reduced puromycylation of Mff and MiD49 in *CLUH* KO cells compared with control cells. This suggests that loss of *CLUH* results in decreased translation of Mff and MiD49. In addition, protein translation of Mff and MiD49 occurs in the cytosol or near mitochondria, but not on the mitochondria (see Fig 8k-p).

5. The physiological significance of the proposed interaction is not clear (at least for the HeLa cells). One possibility is to look at stress conditions in which CLUH levels increase, and check the impact on mitochondria fission (via MiD49 of Mff, through their knock down). Since CLUH was shown to affect mRNAs encoding OXPHOS components, maybe conditions that are related to this will be relevant.

We thank Reviewer #1 for suggesting this interesting line of investigation to explore the physiological significance of our proposed interaction in HeLa cells. It was shown that in mice, mRNA and protein levels of CLUH increase in the liver during fetal development, peak shortly after birth at P0-1, and decrease after weaning. *CLUH* is essential for the metabolic remodeling during the fetal-neonatal transition, and loss of *CLUH* results in neonatal lethality (Schatton *et al.*, *JCB* 2017). However, whether and how mitochondrial morphology changes in these processes were not explored. Currently there are no other known physiological conditions in which CLUH levels increase, but we will continue searching for such conditions, and look forward to addressing this point in the future.

6. For the experiment with Mff and MiD sequestration to the lysosome, it will be informative to present the impact of sequestration on mitochondria morphology. Presumably, mitochondria fission should be affected.

We thank Reviewer #1 for this brilliant idea. We have carried out the experiments accordingly. Indeed, expression of LAMP1-Mff or LAMP1-MiD49 in HeLa cells led to significantly elongated and/or clustered mitochondria, with LAMP1-MiD49 resulting in a stronger phenotype. These are due to a significant portion of Drp1 being sequestered to lysosomes, and a corresponding decrease in the portion of Drp1 being recruited to mitochondria in these cells. In contrast, expression of LAMP1-CLUH in HeLa cells did not result in significant changes of mitochondrial morphology as compared to non-transfected cells, because LAMP1-CLUH does not sequester Drp1 to lysosomes (see Supp Fig 6a-e).

Reviewer #2 (Remarks to the Author):

Clueless (Clu/CLUH) is an RNA-binding protein with important functions in regulating mitochondria morphology and distribution. Drp1 is a key regulator of mitochondrial fission. In this work, the authors interrogate the relationship between Clu and Drp1 using *Drosophila* and

cultured cell models. Clu loss-of-function and over-expression phenotypes are described in the *Drosophila* muscle and additionally in the context of mitochondrial quality-control mutants (i.e., pink and parkin mutants). Gross morphological changes in mitochondria shape and distribution following loss or overexpression of mammalian CLUH are similarly presented. Because overexpression of drp1 partially rescues the clu null phenotype, Clu is suggested to function upstream of Drp1 activity. Consistent with this idea, drp1 over-expression seems to increase Drp1 localization to mitochondria. Co-IP data indicate Clu and Drp1 associate biochemically in *Drosophila* extracts. Further analysis suggests the interaction between CLUH and Drp1 is indirect and may be mediated by OMM receptors, Mff and MiD49. Intriguingly, manipulation of CLUH dosage appears to alter steady state levels of Mff and MiD49 protein, although mechanisms underlying this response remain unclear.

Overall, the authors present a body of data nicely implicating a genetic and biochemical relationship between Drp1 and Clu. The work is of significance given implications to mitochondrial integrity and quality, so essential to health and development. Nevertheless, failure to display quantification of key findings makes it difficult to interpret the work or determine veracity of the stated conclusions. Finally, the relationship between Clu and Mff or MiD49 protein levels remains murky, so the model presented in Figure 8 seems highly speculative. No evidence indicating translational control is presented or discussed beyond changes in protein levels via western blot. I recommend revision prior to publication.

We thank Reviewer #2 for all the constructive comments to improve our manuscript.

Major Points:

1. The authors nicely present quantification of some data (e.g. Supp. Fig 1F and Fig 2G present quantification of mitochondrial size), however it is imperative that the same rigor of quantification is presented throughout the manuscript. I also discourage the use of bar charts for most of these measurements; showing individual data points through scatter plots is recommended as it helps the reader see if the data follow a given trend, if there are significant outliers, and also how many samples were quantified. Quantification is required for the following:

Fig 1 A-L

Fig 3 C-E''

Fig 4 D-F''

Fig 5 C-I'

Fig 6 G-h''

Fig 7 F-H'' and Fig 7 K (Show RNA levels quantified across replicates).

Additionally, please indicate N number of cells examined per experimental replicate for all panels and how many cells displayed a given phenotype.

We appreciate Reviewer #2 for these great suggestions on the utilization of scatter plots for data quantification and presentation. We have added the quantifications using scatter plots, and updated the charts in Figs 1, 3, 4, 5, 6, 7. We also added the number of *Drosophila* tissue samples or the number mammalian cells analyzed for each experiment to the figure legend where appropriate, and added the exact percentages in parentheses where appropriate in the main text.

2. How Clu/CLUH regulates Mff or MiD49 protein levels remains unclear. Further unclear is whether Mff and MiD49 are subject to the same regulatory process, as over-expression of CLUH results in an increase of MiD49 protein levels, while loss of CLUH leads to a corresponding loss of MiD49. The results for Mff are far more subtle, although not discussed by the authors. Statistically, over-expression of CLUH does not alter Mff, while loss of CLUH leads to a

decrease (but not loss) of Mff. These distinctions between Mff and MiD49 responses are likely biologically relevant and should be noted.

We thank Reviewer #2 for raising this suggestion. In response to Reviewer #2, we have added a paragraph (Discussion, Lines 418-429) to discuss “While MiD49 and Mff are both regulated by CLUH, we did observe some differences in terms of the strength of regulation”, in addition to our original Discussion (Lines 410-417) that “Differential regulation of different Drp1 receptors may provide a mechanism for fine-tuning Drp1 recruitment and activity in response to different subcellular stimuli.”

3. What contributes to the decrease of Mff or MiD49 proteins upon loss of CLUH is not addressed. While the authors present in Figure 7K a semi-quantitative RT-PCR gel and 7L shows a protein:RNA ratio, it is necessary to show replicated qPCR data of normalized Mff and MiD49 levels in WT vs CLUH KO vs CLUH OE samples. A protein:RNA ratio is not particularly helpful, as protein production does not necessarily scale linearly with RNA amount. The semi-quantitative gel does not seem to indicate loss or destabilization of Mff or MiD49 mRNAs in response to loss of CLUH; confoundingly, these mRNA levels actually seem higher, but a qPCR would show this more clearly. The authors should then provide interpretation of the RNA vs protein level data and discuss models.

We appreciate Reviewer #2 for these thoughtful comments and have run qPCR experiments accordingly. Replicated qPCR data showed that normalized Mff and MiD49 mRNA levels are not significantly changed in response to either *CLUH* KO or *CLUH* OE, as compared to those in control HeLa cells. We have replaced the RT-PCR panels with the qPCR data (Fig 8h).

We also thank Reviewer #2 for bringing up the nonlinear scale between protein production and mRNA amount. Accordingly, instead of presenting protein:RNA ratio, we used the Puro-PLA assay (also suggested by Reviewer #2 in Major Point #4) to investigate regulation of Mff/MiD49 translation by CLUH (discussed below).

4. The model presented in Figure 8 is speculative. There are various ways one can test translational status, such as by visualizing nascent translation in vivo (e.g. Puro-PLA or Sun-TAGs) or by examining ribosomal occupancy or distributions of proteins in polysome gradients, but these data are not presented here. While translational control is one model to account for reduced steady-state Mff or MiD49 proteins in the absence of CLUH, other models may be considered, such as protein destabilization/degradation.

We thank Reviewer #2 for this excellent suggestion. We have carried out the Puro-PLA assay, and found significantly reduced puromycylation of Mff and MiD49 in *CLUH* KO cells compared with control cells. This suggests that loss of *CLUH* results in decreased translation of Mff and MiD49 (see Fig 8k-p). We have also added in the text (Lines 378-380) “Our results strongly argue for translational control of Mff and MiD49 by CLUH. However, other contributing mechanisms, such as regulation of protein degradation, cannot be completely ruled out.”

Minor Points

1. Statistical analyses are presented throughout, although the authors should reconsider the use of the Student’s t-test when making multiple comparisons in Figs 1, 3B, and 7. Likewise for Supp Fig 1B, 2A, 3K.

We thank Reviewer #2 for the suggestion regarding statistical analysis. We have replaced the Student’s t-test with one-way ANOVA for multiple comparisons in Figs 1, 3b, and 7 (now Fig 8g-h) and Supp Figs 1b, 2a, 3k (now Fig 5k).

2. Rescue of *clu* null phenotypes by OE of *drp1* is convincing, albeit partial. The authors should

note the rescue is partial or incomplete and acknowledge the idea that other factors also function downstream of Clu important for viability.

We agree with Reviewer #2 that the partial rescue could reflect the fact that other factors important for viability, in addition to *drp1*, also function downstream of *clu*. A similar point was raised and has been addressed above (Reviewer #1, Point #2a).

3. Does manipulation of Clu dosage in Drosophila (RNAi, mutant, or OE) alter Mff protein levels?

We would definitely like to investigate Mff protein levels in *Drosophila*, but the lack of *Drosophila* Mff antibodies prevented us from performing these experiments. A similar point has been addressed above (Reviewer #1, Point #2b).

4. The authors use two independent techniques to show that loss of clu does not alter Drp1 protein levels (Fig 4A and D). It was surprising to note in Fig 4D, Drp1 signals seem relatively unaffected regardless of Clu dosage, given, in cultured cell models, the authors suggest that CLUH depletion may lead to a decrease in Drp1 recruitment to mitochondria (Fig 6E). Could the authors discuss this discrepancy?

In both Drosophila and cell lines, CLUH OE seems to increase overlap of mitochondria and Drp1 signals (Fig4F, Fig 6H – although both need quantification, perhaps through a Pearson's coefficient). Is one system more sensitive to Clu/CLUH loss, while both respond comparably to Clu/CLUH OE?

We thank Reviewer #2 for these thoughtful questions. We have looked at Drp1 recruitment to mitochondria in response to loss of *clu* in *Drosophila* more closely. Rather than using ovary, due to the limitation of resolution and tools to co-stain Drp1 and mitochondria efficiently in nurse cells, we used *Drosophila* muscle for this experiment, as we did for *clu* OE (Fig 4f-h"). Results showed that *clu* RNAi resulted in significantly decreased Drp1 recruitment onto mitochondria (Fig. 4h-h"). We used Pearson's coefficient to quantify the overlap of mitochondria and Drp1 signals in both *Drosophila* and HeLa, in response to *clu*/CLUH OE and *clu*/CLUH loss-of-function (see Fig 4i and Fig 6i, o). Together, we concluded that both *Drosophila* and mammalian systems are sensitive to *clu*/CLUH loss.

For Fig 4d-d', after magnifying the images, we did see reduced Drp1 signals at several locations in the peri-nuclear region in *clu* null cells. By contrast, Drp1 was evenly distributed in the cytosol in the neighboring WT cells. Considering the peri-nuclear mitochondrial clustering phenotype in *clu* null cells, these Drp1-absent locations could represent where mitochondria are localized (i.e. decreased overlap of mitochondria and Drp1 signals in *clu* null mutant nurse cells).

5. Through cell fractionation, the authors show CLUH OE leads to an increase in Drp1 levels in the mito fraction and a corresponding loss in the cytoplasm. These are nice data, but can the authors discuss why the recruitment of CLUH itself to the mito fraction seems relatively modest? Could this support the hypothesis where CLUH itself is not directly responsible to recruit or drag Drp1 to mitochondria? (This would be consistent with data shown in Fig 7F–H")

We thank Reviewer #2 for this important observation. Clueless/CLUH are cytosolic proteins, which are not directly localized to mitochondria but occasionally "juxtaposed" to mitochondria in *Drosophila* and mammalian cells (Cox and Spradling, *DMM* 2009; Gao et al., *JCB* 2014). As Reviewer #2 suggested, this indeed supports our hypothesis that CLUH itself is not directly responsible to recruit Drp1 onto mitochondria (Fig 7). Instead, CLUH achieves promoting mitochondrial recruitment of Drp1 through binding Drp1 receptor mRNAs and regulating their translation in the cytosol (see RNA IP and Puro-PLA results in Fig. 8i-j, k-p; also discussed above - Reviewer #2, Major Point #4) and therefore positively regulating Drp1 receptor protein levels (see Fig. 8a-g).

6. For cell culture work, CLUH CRISPR KO lines are compared to "WT" cells. Please define

these controls. Are they non-edited CRISPR lines or other? One must consider potential for off-target effects.

Absolutely, the “WT” cells are non-edited HeLa cells. Also, we have controlled off-target effects by generating two independent *CLUH* KO cell lines (*CLUH-40* and *CLUH-43*, Fig 5a-b), which showed consistent phenotypes including slow growth, smaller cell size and mitochondrial clustering. In addition, we overexpressed *CLUH* by plasmid transfection in both *CLUH* KO HeLa cell lines, which robustly rescued the mitochondrial clustering phenotype in both *CLUH* KO cell lines (Supp Fig 3c-f). This indicates that the clustered mitochondria in these *CLUH* KO cells are indeed due to loss of *CLUH*, rather than off-target effects.

7. In the absence of quantification, statements on lines 227 (“significant reduction”) and 250 (“significantly increased colocalization”) are misleading, as the data are not shown. These metrics need to be quantified before such qualitative statements can be made.

We understand the reviewer’s point. For the statement on Line 227 (previous Fig. 5G-I’, now Fig 5h-j’), quantification of mitochondrial morphology is now shown in Fig 5k. For the statement on Line 250 (previous Fig 6G-h”, now Fig 6k-n”), we have quantified the co-localization between Drp1 and mitochondria using Pearson’s coefficient (see Fig 6o).

8. The data shown in Fig 6 D-F are not particularly compelling. Specifically, how the Drp1 fluorescence intensity at mitochondria was calculated is not clear. The authors may consider a more standard colocalization analysis, such as a Pearson’s or Manders correlation coefficient.

We thank Reviewer #2 for this suggestion. We have quantified the co-localization between Drp1 and mitochondria using Pearson’s coefficient (see Fig. 6i). We have also modified the figure legend of Fig 6 and the “Confocal imaging and image analysis” paragraph (Lines 759-770) in the “Methods” section, to clarify the image processing procedure for measurement of mitochondria-localized Drp1 fluorescence intensity.

9. In the Methods (lines 379-90), please add dilutions of antibodies used for IF vs western blotting. These details are key to ensure reproducibility.

We have added dilutions of antibodies used for IF vs western blotting to the “Antibodies used for IF, WB, IP and PLA” paragraph (Lines 691-714) in the “Methods” section.

10. The TPR domain of Clu is important for RNA-binding: <https://www.ncbi.nlm.nih.gov/pmc/articles/PMC4823986/>

Have the authors examined whether delta TPR mutants recapitulate mutant phenotypes? This line of experimentation would get to the question of mechanism, as related to possible contributions regulating Mff and MiD49.

We appreciate Reviewer #2 for this interesting line of experiment. We have now performed this experiment and results were as expected. We found that expression of full-length *CLUH* results in mitochondrial fragmentation in HeLa cells (Fig 5j-k); however, expression of *CLUH*-delta-TPR does not (Supp Fig 7h-i), indicating that the RNA-binding activity of *CLUH* (i.e. translational regulation of Mff and MiD49 by *CLUH*) is essential for regulation of mitochondrial fission by *CLUH*.

11. Is there evidence for a role for Clu in RNA localization? Are Mff or MiD49 mRNAs localized appropriately in *CLUH* KO or OE sample?

We thank Reviewer #2 for asking this question. We have used RNAscope to visualize localizations of *Mff* and *MiD49* mRNAs, and found no significant changes in the localization of *MiD49* or

Mff mRNAs in response to either *CLUH* OE or *CLUH* KO (Supp Fig 7a-g). We concluded that *CLUH* regulates *Mff* and *MiD49* protein levels through translational control (via Puro-PLA, Fig 8k-p), rather than transcriptional regulation (via qPCR, Fig 8h) or regulation of mRNA localization (via RNAscope, Supp Fig 7a-g).

Reviewer #3 (Remarks to the Author):

The manuscript by Yang H et al. entitled "Clueless/CLUH regulates mitochondrial fission by promoting recruitment of Drp1 to mitochondria" describes the role of the RNA-binding protein *CLUH* in Drp1-dependent mitochondrial fission using *Drosophila* and mammalian cell culture as two model systems.

Strengths:

- The authors convincingly demonstrate the pivotal role of *CLUH* for mitochondrial morphology and mitochondrial function with implications for fitness, in their two model systems
- The authors provide a number of convincing pieces of evidence that *CLUH* functions on mitochondrion-localized Drp1, which enables mitochondrial fission

Limitations:

- Although the authors provide data suggesting that *CLUH* is involved in recruiting Drp1 to mitochondria, these data are comparatively weak, and the proposed mechanisms behind requires experimental validation (see below)
- The authors ignore the potential role of peroxisomes, which share parts of the fission machinery with mitochondria

We thank Reviewer #3 for all the constructive comments to improve our manuscript.

Major points:

- 1.) The *CLUH*-dependent recruitment of Drp1 to mitochondria needs further experimental validation or the interpretation of existing data must be handled with more caution. The authors should perform at least two of the suggestions (or equivalent approaches) to substantiate their model (preferentially as described in d):
 - a.) Fig. 6D-F: The effects on mitochondrial Drp1 localization for *CLUH* ko are very moderate, I recommend cell fractionation experiments as shown in Fig. 6I with *CLUH* ko cells.
 - b.) Fig. 6G-I: The effects on mitochondrial Drp1 localization for *CLUH* OE appear substantial, but then fluorescence microscopy experiments as in Fig. 6D-F should result in similar effects. I recommend performing these experiments.
 - c.) Fig. 6: I recommend expressing *CLUH* in *CLUH* ko cells using a controllable promoter. Drp1-recruitment which depends on the expression level of *CLUH* would be more convincing. Alternatively, time-dependent recruitment of Drp1 upon *CLUH* expression using live-cell microscopy could be used.
 - d.) Fig. 7KL: The authors show that the steady state levels of the mRNA encoding the Drp1 receptors *MiD48* and *Mff* are decreased in the absence of *CLUH*. However, it remains unclear whether *CLUH* binds and stabilizes these mRNAs in this context, and most importantly, it remains unclear whether this binding occurs at the OMM. Any experimental validation of the latter would help a lot to support the model provided by the authors.

We appreciate Reviewer #3 for these thoughtful suggestions. Considering the availability of reagents and resources, we have addressed b.) and d.) accordingly:

b.) We have quantified the co-localization of mitochondria and Drp1 signals using Pearson's coefficient for both *CLUH* KO (now Fig 6e-h") and *CLUH* OE (now Fig 6k-n") panels. *CLUH* KO resulted in significant decreases in recruitment of Drp1 to mitochondria (Fig 6i), and *CLUH* OE led to significantly increased recruitment of Drp1 to mitochondria (Fig 6o), as compared to WT cells. Also, we

have modified the format of *CLUH* KO panels (Fig 6e-h’), so that they match that of the *CLUH* OE panels (Fig 6k-n’).

d.) We performed RNA IP (RIP) experiments, which showed that *CLUH* binds mRNAs of Drp1 receptors Mff and MiD49 (Fig 8i-j). We also performed RNAscope experiments, and found Mff and MiD49 mRNAs did not show significant co-localization with mitochondria (Supp Fig 7a-g). In addition, we carried out Puro-PLA assay, and found significantly reduced puromycylation of Mff and MiD49 in *CLUH* KO cells compared with control cells, indicating decreased translation of Mff and MiD49 in response to loss of *CLUH* (Fig8k-l, m-n, quantified in o). Our Puro-PLA results also showed that translation of Mff and MiD49 occurs in the cytosol or near mitochondria, but does not exactly co-localize with mitochondria (Fig8k’-l’, m’-n’, quantified in p). Based on the above evidence, we conclude that *CLUH* binds mRNAs of Mff and MiD49, and promotes their translation, which happens in the cytosol instead of on the OMM.

2.) The model combines interpretations of data from flies and cell culture, and it is not clear which part of the model is validated by fly data only, by fly and cell culture data, and by cell culture data only. This should be indicated in the figure and the figure legend.

We thank Reviewer #3 for bringing up this important point. We have updated the figure legend of the model figure (now Fig 9) accordingly.

3.) The authors need to discuss the potential role of peroxisomes. Could *CLUH* promote Drp1-dependent fission of peroxisomes as well? The reviewer does not request experimental validation but peroxisomes should be at least mentioned in the discussion.

We thank Reviewer #3 for suggesting this interesting line of investigation. We investigated peroxisome morphology (anti-PMP70) in control vs *CLUH* KO HeLa cells, and found no significant morphological changes of peroxisomes when comparing *CLUH* KO to control HeLa cells (Supp Fig 8). This indicates that *CLUH* regulates Drp1-dependent fission of mitochondria, but not that of peroxisomes.

We have also added to the text (Discussion) to clarify the issue (Lines 430-434): “Drp1 and Mff have also been shown to regulate peroxisome fission. However, we did not observe significant morphological changes of peroxisomes in *CLUH* KO cells as compared to control cells (Supplementary Figure 8), despite decreased Mff levels and reduced recruitment of Drp1 to mitochondria in *CLUH* KO cells. This indicates that *CLUH* may only regulate mitochondria-localized Mff and recruitment of Drp1 to mitochondria, but not peroxisome-localized Mff or Drp1.”

Minor points:

1.) Fig. 5: Quantifications for changes in mitochondrial morphologies are recommended.

We understand the reviewer’s point. Accordingly, quantifications for changes in mitochondrial morphologies are now shown in Fig 5g, k, p.

Finally, we would like to thank the reviewers for their time, effort, and excellent comments and suggestions, which we believe have greatly clarified various points and strengthened our manuscript.

REVIEWERS' COMMENTS

Reviewer #1 (Remarks to the Author):

The authors addressed all my concerns in this revised manuscript. The additional data and analysis significantly strengthen their conclusions.

One comment: Impact of CLUH depletion on mRNA localization to human mitochondria was tested previously (Vardi-Oknin and Arava (2019)). This should be referenced, too.

Reviewer #2 (Remarks to the Author):

Yang et al. present a revised manuscript detailing a requirement for Clueless (Clu) to regulate Drp1, required for mitochondrial integrity. Through new data, they identify an RNA-protein complex between Clu and the Drp1 receptors Mff and MiD49. They propose, and now present evidence to support, a model whereby Clu promotes the translational activation of Mff and MiD49, as shown with Puro-PLA. They also nicely complement these data by showing overexpression of a mutant form of Clu lacking RNA-binding motifs (delta TPR) bypassed the mitochondrial phenotypes. Most importantly, the phenotypes presented in the revised study are rigorously quantified with appropriate statistical tests.

I commend the authors for such a diligent job addressing the revisions. They have addressed all of the points raised in my prior critique. As a result, the work is much more solid and I support its publication.

Reviewer #3 (Remarks to the Author):

The authors adequately addressed all my points. Therefore, I recommend publication of the manuscript.

Point-by-point response to the reviewers' comments

Reviewer #1 (Remarks to the Author):

The authors addressed all my concerns in this revised manuscript. The additional data and analysis significantly strengthen their conclusions.

One comment: Impact of CLUH depletion on mRNA localization to human mitochondria was tested previously (Vardi-Oknin and Arava (2019)). This should be referenced, too.

We thank Reviewer #1 for the positive feedback and the suggestion. We have added Vardi-Oknin and Arava (2019) to the References (as Reference #43), and cited this paper in the following sections of our manuscript:

1. Introduction (Lines 66-67): “Clu orthologues have been identified as RNA-binding proteins in yeast [41], *Drosophila* [42] and mammalian cells [39, 43].”
2. Results - *CLUH* regulates MiD49 and Mff protein levels in mammalian cells (Lines 354-356): “*CLUH* has been shown to bind nuclear-encoded, mitochondria-targeted mRNAs and regulate their localized translation near mitochondria [39, 43, 45]. To examine if *Mff* and *MiD49* mRNAs are targets bound by *CLUH*, we performed RNA Immunoprecipitation (RIP).”
3. Discussion (Lines 424-426): “*CLUH* has been shown to bind multiple mRNAs of nuclear-encoded, mitochondria-destined proteins, and regulate translation and stability of these mRNAs [39, 42, 43, 45].”

Reviewer #2 (Remarks to the Author):

Yang et al. present a revised manuscript detailing a requirement for Clueless (Clu) to regulate Drp1, required for mitochondrial integrity. Through new data, they identify an RNA-protein complex between Clu and the Drp1 receptors Mff and MiD49. They propose, and now present evidence to support, a model whereby Clu promotes the translational activation of Mff and MiD49, as shown with Puro-PLA. They also nicely complement these data by showing overexpression of a mutant form of Clu lacking RNA-binding motifs (delta TPR) bypassed the mitochondrial phenotypes. Most importantly, the phenotypes presented in the revised study are rigorously quantified with appropriate statistical tests.

I commend the authors for such a diligent job addressing the revisions. They have addressed all of the points raised in my prior critique. As a result, the work is much more solid and I support its publication.

We thank Reviewer #2 for the positive feedback.

Reviewer #3 (Remarks to the Author):

The authors adequately addressed all my points. Therefore, I recommend publication of the manuscript.

We thank Reviewer #3 for the positive feedback.

Finally, we thank all the reviewers for their insightful comments and suggestions, which have greatly strengthened our manuscript.